# Prospects for improving the representation of coastal and shelf seas in global ocean models

Jason Holt[1], Pat Hyder[2], Mike Ashworth[3], James Harle[1], Helene T. Hewitt[2], Hedong Liu[1], Adrian L. New[4], Stephen Pickles[3], Andrew Porter[3] , Ekaterina Popova[4], J. Icarus Allen[5], John Siddorn[2], Richard Wood[2]

[1]National Oceanography Centre, 6 Brownlow Street, Liverpool, L3 5DA, UK

[2]Met Office Hadley Centre, FitzRoy Rd, Exeter, EX1 3PB, UK.

[3]STFC Daresbury Laboratory, Sci-Tech Daresbury, Warrington, WA4 4AD

[4]National Oceanography Centre, European Way, Southampton, SO14 3ZH, UK

[5]Plymouth Marine Laboratory, Prospect Place, Plymouth, PL1 3DH, UK

*Correspondence to*: Jason Holt (jholt@noc.ac.uk)

**Abstract.** Accurately representing coastal and shelf seas in global ocean models represents one of the grand challenges of Earth System science. They are regions of immense societal importance, through the goods and services they provide, hazards they pose and through their role in global scale processes and cycles, e.g. carbon fluxes and dense water formation. However, they are poorly represented in the current generation of global ocean models. In this contribution we aim to briefly characterise the problem, and then to identify the important physical processes, and their scales, needed to address this issue in the context of the options available to resolve these scales globally and the evolving computational landscape.

We find barotropic and topographic scales are well resolved by the current state-of-the-art model resolutions, e.g. nominal 1/12°, and still reasonably well resolved at 1/4°; here the focus is on process representation.  We identify tides, vertical coordinates, river inflows and mixing schemes as four areas where modelling approaches can readily be transferred from regional to global modelling with substantial benefit. In terms of finer scale processes, we find that a 1/12° global model resolves the 1st baroclinic Rossby Radius for only ~8% of regions <500m deep, but this increases to ~70% for a 1/72° model, so resolving scales globally requires substantially finer resolution than the current state-of-the-art.

We quantify the benefit of improved resolution and process representation using 1/12° global and basin scale northern North Atlantic NEMO simulations; the latter includes tides and a k-ε vertical mixing scheme. These are compared with global stratification observations and 19 models from CMIP5. In terms of correlation and basin wide RMS error, the high resolution models out perform all these CMIP5 models. The model with tides shows improve seasonal cycles compared to the high resolution model without tides. The benefits of resolution are particularly apparent in Eastern Boundary upwelling zones.

To explore the balance between the size of a globally refined model and that of multiscale modelling options (e.g. finite element, finite volume or a 2-way nesting approach) we consider a simple scale analysis and a conceptual grid refining approach. We put this analysis in the context of evolving computer systems, discussing model turn-around time, scalability and resource costs.  Using a simple cost-model compared to a reference configuration (taken to be a 1/4° global model in 2011) and the increasing performance of the UK Research Councils' computer facility, we estimate an unstructured mesh multiscale approach resolving process scales down to 1.5km would use a comparable share of the computer resource by 2021, the 2-way nested multiscale approach by 2022, and a 1/72° global model by 2026. However, we also note that a 1/12° global model would not have a comparable computational cost to a 1° global model today until 2027. Hence, we conclude that for computationally expensive models (e.g. for oceanographic research or operational oceanography), resolving scales to ~1.5km would be routinely practical in about a decade given substantial effort on numerical and computational development. For complex Earth System Models this extends to about two decades, suggesting the focus here needs to be on improved process parameterisation to meet these challenges.

## 1. Introduction

Improving the representation of coastal and shelf seas in global models is one of the grand challenges in ocean modelling and Earth System science. Global ocean models often have poor representation of coastal and shelf seas (Renner et al., 2009;Holt et al., 2010), further quantified below, due to both their coarse resolution and their lack of coastal-ocean process representation. See Griffies and Treguier (2013) for a recent review of the state of the art in global ocean modelling. In this paper we aim to identify the relevant physical processes, quantify the horizontal scales needed to resolve these processes and explore the approaches that could be employed to realise an improvement. In particular we compare the relative merits of a continued refinement of quasi-uniform structured grids with multiscale approaches, which would allow increased resolution where required. The multiscale approach could, for example, use unstructured meshes or multiple two-way nested grids. There have been other previous explorations of the scales important in shelf sea models (Greenberg et al., 2007;Legrand et al., 2007). These have tended to focus on specific numerical methods and approaches, largely around triangular unstructured meshes. Here we step back from a detailed analysis of the numerics and consider, in general terms, what is likely to be practical to achieve improved coastal and shelf sea modelling on a global scale, on what time scales and what the ways forward may be. We primarily draw on experience with the NEMO model (Nucleus for a European Model of the Ocean; Madec, 2008) to provide a specific context, but expect the conclusions drawn to be generic.

The remainder of this section describes the background and motivation. Coastal-ocean processes and scales and their relation to global quasi-uniform model grids are described in section 2. Section 3 considers modelling approaches that might address coastal-ocean process representation and resolution, drawing on the CMIP5 coupled ocean-atmosphere climate models (Taylor et al., 2012) and two 1/12° NEMO configurations in comparison with EN4 profile observations (Good et al., 2013) to provide quantitative examples. These considerations are related to changing computer architectures and issues of model performance in section 4, to estimate when they may be practical. The paper ends with conclusions in section 5.

### 1.1 Background and Motivation

Coastal and shelf seas represent a small fraction of the area of the global ocean (9.7% of the global ocean is <500m deep and 7.6% <200m), but have a disproportionately large impact on many aspects of the marine environment and human activities. While, our focus here is on modelling physical-ocean processes, facets of marine biogeochemistry and ecosystems, and the climate system often provide the underlying motivation. These seas are the most highly productive regions of the world ocean, providing a diverse range of resources (e.g. food, renewable energy, transport) and services (e.g. carbon and nutrient cycling and biodiversity), and also expose human activity to hazards such as flooding and coastal erosion.

The geography of these seas is very varied including semi-enclosed seas, broad open shelves, narrow shelves exposed to the open ocean, and coastal seas behind barrier islands. Rather than adopt a typological approach (e.g. Liu et al., 2010) we focus on generic physical processes described by some straightforward spatially-varying properties, as is appropriate for the global case; regional model studies would go beyond this to consider the detailed conditions specific to the region and tailor the model accordingly. While many of the largest shelf seas are in polar regions, we limit our investigation here to liquid water modelling and leave considerations of sea-ice modelling in this context to further work.

The study of coastal and shelf seas in a global context involves both upscaling (small scales influencing large) and downscaling (large scales influencing small) considerations, alongside the internal dynamics. Both dynamics and biogeochemistry provide motivations to studying the influence of coastal-ocean processes on a global scale. A particularly important dynamical feature is the formation of dense water on Arctic and Antarctic (Orsi, 2010;Orsi et al., 1999) shelves and its subsequent downslope transport and mixing to form deep water masses through the "cascading" process, thereby contributing to the global thermohaline circulation. Similarly coastal-upwelling is both an important control of air-sea heat flux with implications for regional climate (e.g. in the southeast Pacific; Lin, 2007) and a key process in global marine ecosystems. The coastal-ocean plays a key role in global biogeochemical cycles, for example through the drawdown of carbon in highly productive shelf seas

and its transport either to on-shelf sediments or off-shelf to the deep ocean, where it is isolated from atmospheric exchange (Bauer et al., 2013;Chen and Borges, 2009). Shelf seas are also a source of potent greenhouse gases, such as nitrous oxide (Seitzinger and Kroeze, 1998) and methane release from hydrates (Shakhova et al., 2010). The coastal-ocean is the first point of entry for all material of terrestrial origin entering the marine environment, for example freshwater from rivers and ice

sheets/shelves, inorganic nutrients, organic material and anthropogenic pollutants and this material can be substantially modified as it is transported across the coastal-ocean (Barrón and Duarte, 2015). Hence the coastal- and open-ocean biogeochemical cycles are intimately coupled. There is still substantial uncertainty in their role and feedbacks with the wider climate system, and making progress on this is largely dependent on the accurate simulation of the physical environment in the coupled coastal-open-ocean system.

Investigating the large scale impacts on smaller scale processes in the coastal-ocean can often be successfully treated by (one-way or two-way) nested regional studies, focusing on an area of interest ranging from local (e.g. Zhang et al., 2009) to regional (e.g. Wakelin et al., 2009) to basin (e.g. Holt et al., 2014;Curchitser et al., 2005) scale. There are, however, occasions where a global or quasi-global approach is appropriate. These relate to cases where it is important to consider impacts on human systems of global relevance. Examples include global food security and the role of Living Marine Resources in ensuring this

(Merino et al., 2012;Barange et al., 2014), and quantifying global vulnerability to sea level rise and coastal-flooding (e.g. Nicholls, 2004). Moreover, cases where basin scale oceanic processes directly influence the coastal-ocean are best considered on a global scale (Popova et al., 2016), as regional simulations may be compromised by errors propagating from simplified boundary condition approaches (see below). Coastal upwelling (Popova et al., 2016; Hobday and Pecl, 2014) and impacts of changes in western boundary currents (Wu et al., 2012) are notable examples.

While regional or local models often provide the optimal solution for many coastal-ocean questions there is a significant overhead in their deployment. A global model with improved representation of the coastal-ocean opens up the opportunity to provide rapid and cost effective information in a particular region for either scientific or operational use, without needing to configure a new domain. A particular example, here is the European Copernicus Service (marine.copernicus.eu). In this multi-€m investment, operational forecast and reanalysis products are provided to a range of users, from bespoke models of several

European regions. If the global model in this service had improved coastal-ocean representation then a similar, but not optimal, range of information could be provided for a much wider range applications around the world, notably where this level of investment is not available.

Hence we define the context of the present study to be the improvement of the representation of the coastal-oceans in four classes/uses of global ocean models: i. global climate models, ii. global earth systems models, iii global models used as a

resource for regional scientific studies, and iv. global models providing regional operational information.

### 2. Coastal-Ocean processes and scales

The distinct physical characteristics of the coastal-ocean, in comparison to other oceanic regions, are largely determined by their shallow depth and proximity to land. This has several implications for the dynamics:

- The water depth is generally similar to or not much greater than the surface or seabed boundary layers, so

35        turbulence and mixing is invariably important.

- Extreme variations in topography (compared with the water depth) are a defining feature.
- Incident waves grow in amplitude in shoaling water to conserve energy flux, so, for example, these can be regions of large tides.
- The inertia (thermal and mechanical) of shelf seas is small, so they are highly constrained by external forcing.

- The horizontal length scales of the dominant physical processes decrease with decreasing depth (see below) and so are generally much smaller than in the deep ocean.

- Rivers and glacial melt provide a source of buoyant fresher water that forms coastal currents and impacts stratification and mixing near the coast.
- In Polar Regions, land provides both a point of attachment (land fast ice) and a source of divergence (polynias) for sea ice.

Alongside these internal dynamics, coastal-open-ocean coupling is of critical importance to the considerations here. At ocean margins currents tend to follow contours of the Coriolis parameter divided by water depth ($f/h$), and so coastal regions are largely isolated from the large scale geostrophic circulation. Physical processes at the shelf break mediate the transfer of material across this barrier (Huthnance, 1995), e.g. the Ekman drain within the bottom boundary layer, eddies and internal waves; these tend to be fine scale and high frequency.

While there are numerous physical processes at work in shelf and coastal seas the underlying principles and equations are the same as in the open ocean, and many features noted above are represented in the current generation of global models. Their relative importance and scale differ significantly in the two cases, and so does how they are treated in numerical models. The processes are reviewed by Robinson and Brink (1998), Huthnance (1995) and Holt et al (In Press), so we do not discuss the dynamics in any detail here; we are primarily concerned with their characteristic horizontal scales (Table 1).

Ocean tides make a substantial contribution to the mixing and transport in most coastal-ocean regions. For example, the mean $M_2$ semi-major axis tidal current speed is 0.29ms$^{-1}$ for water shallower than 500m, compared with a global mean of 0.06ms$^{-1}$ (based on TPX data; Egbert and Erofeeva 2002). Only 8% by area of these shallow regions have tides <0.12ms$^{-1}$ (i.e. weak tides). The barotropic tide propagates on-shelf as a coastal-trapped wave (CTW), amplifying, and transferring energy to higher harmonics as the water depth shoals; the scale ($L_{bt}$) is characterised by either their wave length or Kelvin wave scale. The
(substantially finer) scale of rectification of tidal currents around topography and the periodic mixing and stratification at fronts is set by the tidal excursion ($L_e$, e.g. Polton, 2014). Topographic steering of currents is a characteristic feature of shelf seas and ocean margins (e.g. the Dooley Current in the North Sea), with a barotropic scale of the water depth divided by the slope ($L_T$) (Greenberg et al., 2007). Other topographic scales, such as the size of individual features, will be locally relevant, but are not considered here.

The annual stratification cycle is a key feature of many shelf seas that are shallower than the winter ambient open-ocean mixed-layer depth. This is well described by a balance between surface heating and mixing (Simpson and Hunter, 1974) and the general spatial pattern is then set by the propagation of tides across the shelf and the topography (i.e. the barotropic scales; $L_{bt}$, $L_T$). Mixed and seasonally stratified waters are bounded by sharp mixing fronts. These provide effective barriers to lateral transport, and drive baroclinic frontal jets (Hill et al., 2008), at a scale characterised by the 1$^{st}$ baroclinic Rossby Radius ($L_1$;
Table 1). While mesoscale eddies are present in shelf seas (e.g. Badin et al., 2009) their importance in dynamics and transport on-shelf is much less clear than in the open-ocean (Hecht and Smith, 2008) or for ocean-shelf transport (e.g. Zhang and Gawarkiewicz, 2015). Coastal upwelling, and consequent frontal jets and filaments (Peliz et al., 2002) also scale with the Rossby Radius.

     Tidal flow over topography in a density stratified ocean excites internal waves at tidal frequencies (Baines, 1982), and their
role in mixing at the shelf break (Rippeth and Inall, 2002) and in the vicinity of banks (Palmer et al., 2013) is now well established. Much of the energy resides (at least initially) in the 1$^{st}$ mode, so their scale ($L_{iw}$; Table 1) is closely related to $L_1$. Hence, we see that resolving $L_{bt}$, $L_T$ and $L_1$, is crucial for a wide range of coastal-process representation.

     Riverine and glacial freshwater inputs form buoyant coastal currents that can form a substantial part of the coastal-ocean circulation and an important control mediating the transport of terrestrial material, notably by inhibiting its direct off-shore
transport. Their scale is difficult to quantify in general terms on a theoretical basis. To characterise how well riverine coastal currents are modelled, we consider the minimum of two scales ($L_r$): the horizontal scale characteristic of seabed frontal

trapping, defined as the depth of trapping (Yankovsky and Chapman, 1997) divided by the local slope and the inflow Rossby Radius (Avicola and Huq, 2002) (Table 1).

## 2.1 Coastal-ocean process scales in a global context

To put the scales described above and listed in Table 1 into a global context we calculate values using the global ORCA12 NEMO model (set up by the DRAKKAR group e.g. Marzocchi et al., 2015;Duchez et al., 2014) as a reference grid and bathymetry. This is a tri-polar grid with a coarsest resolution of 9.3km, but decreasing to minimum values of 1.8km in the southern ocean and 1.3km in the Canadian archipelago. The median scale is 6.3km. The bathymetry is a combination of GEBCO and ETOPO2. The process scales are themselves very much dependent on the scale of the information used to calculate them (e.g. the level of detail in topographic roughness used in calculating $L_T$), so a high resolution model grid used in practice is a good starting point, although the results presented below are not generally dependent on this grid choice. Figure 1 shows values of the barotropic ($L_{bt}$) and 1$^{st}$ baroclinic Rossby Radii ($L_1$), the topographic length scale ($L_T$) and the tidal excursion ($L_e$); see figure caption for further details of the calculation.

The barotropic Rossby radius is, as expected, large (>1000km) even at high latitudes, except in shelf seas and near the coast e.g. in 20m water depth at mid-latitudes, $L_{bt}$~100km. For $L_T$, values <100km are widely distributed across the ocean, reflecting features such as ridges and sills. Values <10km are, however, restricted to the slopes at the ocean margins between the deep ocean and either the continents or the continental shelves. For the baroclinic Rossby radius ($L_1$), values <10km occur in high latitude oceans, whereas values <6km are limited to continental shelves. The tidal excursion ($L_e$) is much smaller, generally <10km. It shows an opposite pattern to the baroclinic Rossby radius, being largest at the coast. Where it is very small (e.g. in the open ocean) so is the tidal velocity and it is of minimal importance. Where it is large, however, it can make a significant contribution to local water column mixing/stability and fine scale residual transport.

To assess how model resolution compares with these scales we define a parameter:

$$e=L_x/(\max(\Delta x,\Delta y).E) \hspace{4cm} (1)$$

at each model grid cell (size $\Delta x$, $\Delta y$) of the ORCA12 mesh, i.e. the number of cells per length scale for process '$x$'. We multiply the size of each cell of the original grid ($\Delta x$, $\Delta y$) by a factor, $E$, to approximate other grid resolutions (without needing to generate the grids; e.g. $E=3$ for a nominal 1/4° resolution). We focus on the barotropic and baroclinic Rossby radii and the topographic scale. We do not consider the tidal excursion further in this context, as resolving it is only beneficial in regions where the tide is large. Here we consider the nominal model resolutions listed in Table 2, along with example applications for the global and coastal-ocean cases. It is worth noting the current generation of forced, high resolution global models are of similar resolution to many historic and on-going shelf sea simulations (see references in Table 2). The cumulative distributions of $e$ (Figure 2), weighted by the area of each grid cell, then show the fraction of the model area at a particular resolution that resolves scale $L_x$ with $e$ or more grid cells. This figure also shows the distribution calculated just for water depth <500m i.e. the coastal-ocean. What constitutes adequate resolution then depends on the process in question. Hallberg et al (2013) suggest 2 grid cells per baroclinic Rossby radius gives a good representation of eddy fluxes, so we take $e>2$ to be eddy resolving. If eddies have a characteristic size of ~$2L_1$ (i.e. ½ the wavelength of the fast growing baroclinically unstable mode; Pedlosky, 1987) then we take $e<1$ to be eddy excluding (i.e., a full parameterisation of eddy effects would need to be included in the model) and between these to be eddy permitting. For the barotropic Rossby radius ($L_{bt}$) we take the limits on excluding and resolving to be $e<2$ and $e>10$, on the basis that this scale needs to be well resolved to capture many coastal-ocean processes (as discussed above). For the topographic scale we set the limits at 1 and 3 respectively, since at least three cells are required to represent a topographically constrained jet.

We can therefore demonstrate that a 1/4° global model is eddy resolving for 27% of the globe; this increases to 52% for 1/12°, 77% for 1/36° and 91% for 1/72°. The fraction of the coastal-ocean that is eddy resolving is significantly less: ~8% at 1/12°; 34% at 1/36°; a 1/72° resolution is needed to be eddy resolving over ~70% of the coastal-ocean. The topographic scale is much

more promising. A 1/12° model has $e>3$ for~90% of the global and ~70% coastal-ocean case. Resolving the barotropic Rossby radius is a somewhat more stringent criterion than resolving the topographic scale in the coastal-ocean at 1/4° or coarser resolution.

To explore the ability of models of different resolution to represent river plumes on a global scale, Figure 3 shows the cumulative distribution of the number of rivers (out of the 925 largest by volume flux; Dai et al., 2009) where the scale, $L_r$, is resolved to level $e$. This suggests modelling riverine coastal currents is an extreme challenge for structured grid global models. Using the same criteria limits as for $L_T$, at 1/12° only 38 of the largest 925 rivers meet the 'permitting' criteria. This implies that, while the fresh water balance is correct, its dispersion and transport properties will be limited. This number increases to 165 at 1/72°.

We see from this scale analysis that 1/72° (~1.5km) might be taken as a good target for resolving many small scale coastal-ocean processes such as eddies, upwelling and the largest river plumes. We would also expect it to be adequate for resolving tidal excursions (where important) and internal tides. However, it is important to consider these results in the context of coastal-ocean dynamics and previous modelling experience. Very few regional coastal-ocean modelling studies have been conducted at eddy permitting resolution, yet significant progress in our understanding of the dynamics of these regions has still been achieved. Hence, while 1.5km might be seen as an aspiration, the practicalities of being eddy resolving on-shelf (when/how this can be reached are discussed below) should not be seen as a particular obstacle to making shorter term progress in modelling the coastal-ocean on a global scale, for example by using 1/36° as a compromise resolution (as in many coastal-ocean studies; e.g. Maraldi, et al 2012) or focusing on process representation (e.g. Luneva et al., 2015). Some features with scales of the Rossby radius, such as coastal upwelling, river plumes and frontal jets will still be present in models that do not resolve this scale, they will just not be particularly well represented. For example, continuity will lead to upwelling in a model of any resolution; its horizontal scale will be determined by the grid and numerics rather than the physics. Internal waves and eddies, in contrast, will simply be absent, and so need to be parameterised. The barotropic and topographic scales are vitally important for the accurate modelling of coastal-ocean dynamics, but can be reached at more modest global resolutions.

### 3. The Modelling Approaches

Here we consider, in general terms, how the processes considered above are *represented* by the model dynamical equations or specific parameterisations, and those that are *resolved* by the model grid. The inadequacy of global climate models in the coastal-ocean is frequently quoted but rarely quantified. So we start this section with a consideration of how well the CMIP5 generation of climate models (Taylor et al., 2012) performs in these regions. We focus on the potential energy anomaly (PEA) as a useful measure of water column stratification. The PEA is defined by:

$$\phi = -\frac{g}{h}\int_{z=-h}^{0} z(\rho(T,S) - \rho(\overline{T},\overline{S}))dz \qquad (2)$$

where $h$ is the water depth (here the integration is limited to 200m), g is the gravitational acceleration, ρ the density and z the (positive up-wards) vertical coordinate. An overbar indicates an average over the same depth as the integration. This represents the energy (per depth) needed to mix the water column. It is a commonly used metric for stratification since it is an integral quantity that does not relate to a particular vertical structure or threshold and connects with simple theories of stratification evolution (Sharples and Simpson, 1996; Simpson and Hunter, 1974). Using the historical period (1970-2005) of 19 CMIP5 models, we calculate mean PEA for each month and average over these 36 years to give a mean annual cycle, interpolated onto the Northern North Atlantic 1/12° NEMO Model grid (see below). These models where selected because they all simulate aspects of marine biogeochemistry. We also calculate the PEA for each profile in the EN4 CTD profile dataset (Good et al., 2013), and average these onto the same model grid to give an observed mean annual cycle on a common grid. The model and observed values are then compared to give RMS error (RMSE) and correlation statistics, here mixing spatial and seasonal variability. Figure 4 shows these values calculated for the whole northern North Atlantic and only where depths are less than

500m (approximately the coastal ocean). This shows the general picture that the performance of these models is degraded in the coastal-ocean (RMS errors are higher, correlations are lower). This is the case for all models for RMSE except one (marginally), and 11 out of 19 for correlation; for the models lying above the line, the correlations are either small or they sit very close to the line. This figure also shows that all the higher resolution models (0.5° or finer) perform well, but there is not a clear resolution dependence, e.g. some coarser resolution model perform similarly well.

### 3.1 Process representation

The representation of coastal processes in global ocean models is straightforward, at least in concept. For example, the NEMO model from V3.2 onwards has the capability of simulating both open ocean and shelf sea cases (O'Dea et al., 2012;Maraldi et al., 2012), with capability improving in later versions. This essentially allows these processes to be included by configuration selection as the global model resolution is refined. The open question is whether features pertinent to the coastal-ocean can be introduced into global models without degrading the solution in the open ocean or significantly increasing their computational cost. The model development process is largely focussed around reconciling the differences between coastal-ocean and global ocean approaches; a good guiding principle could well be to minimise the changes needed in the global modelling approach, on the basis that these choices are well suited for the majority of open ocean processes.

#### 3.1.1 Tides

The representation of tides in global models is the natural starting point. There are two approaches that can be considered: direct simulation and parameterisation. Along with tide generating forces, self-attraction, loading and solid earth tides need to be represented to achieve an accurate tidal simulation (e.g. Stepanov and Hughes, 2004). In addition the correct energy dissipation through bottom friction and internal tide generation is required. Baroclinic global tidal models with prognostic temperature and salinity (e.g. HYCOM; Arbic et al., 2012) can directly simulate the internal tide field. However, low mode internal tides can propagate large distances from their generation region, making their impact (e.g. on mixing) hard to adequately parameterise (Simmons et al., 2004). As identified above, global models at resolutions finer than ~1/4° permit low mode internal tides in the open ocean, but not higher modes or wave numbers or internal tides in the coastal-ocean. For example, Niwa and Hibiya (2011) find a strong resolution dependence of barotropic to baroclinic tidal energy conversion with no convergence even at 1/15°. Hence, some form of wave drag parameterisation may be required. Arbic et al (2012) found a carefully tuned wave drag parameterisation is necessary to accurately simulate tides in the isopycnal HYCOM model, whereas Muller et al (2010) found that a wave drag scheme was not required in the geopotential (z-) level MPI model.

Introducing tides into a global models requires changes to a number of model formulations. For example: The accurate representation of the bottom boundary layer by fine near bed vertical resolution, e.g. through terrain (s-) following or Arbitrary Lagrangian Eulerian coordinates (Petersen et al., 2015); turbulence models suited for multiple boundary layers (Burchard et al., 2008); a sophisticated representation of bottom friction, e.g. quadratic friction with log layer formulation (Blumberg and Mellor, 1987) and a semi-implicit bed stress implementation for numerical stability, given the large stresses and thin vertical layers. For conservation reasons (Campin et al., 2004), most global ocean models are now moving towards using a non-linear free surface, as is also required to represent large tidal amplitudes. Tidal dynamics are most accurately represented with a mode-split time stepping approach (e.g. Shchepetkin and McWilliams, 2005) rather than a fully implicit solution; this is also a trend in recent global ocean model development.

The high frequency cross-coordinate surface vertical displacement of isopycnals arising from an energetic internal tide field in a s- or z- coordinate model, but not in an isopycnal model, might be expected to lead to increased spurious mixing, unless accompanied by methods to control it. For example, this motivated the development of the z-tilde coordinate in NEMO (Leclair and Madec, 2011). A recent review of spurious numerical mixing, focusing on global ocean models with energetic ocean eddies, is provided by Griffies and Treguier (2013). With energetic eddying or tidal flow, the non-linear advection of

momentum becomes more important, which poses a challenge for the numerical methods of momentum advection. This ultimately results in spurious dianeutral tracer transports since, even with very accurate tracer advection schemes (Colella and Woodward, 1984;Prather, 1986) or adaptive vertical coordinate systems (Leclair and Madec, 2011;Gräwe et al., 2015), this will produce spurious transport and/or dispersion errors if the velocity field contains too much energy near the grid scale (Ilicak et al., 2012). They show the spurious dianeutral transport is proportional to the 'Grid-scale Reynolds number', defined as $Re_\Delta=min(\Delta x, \Delta y)U/K_H$, where $K_H$ is the Laplacian viscosity that dissipates the mechanical energy. $Re_\Delta$ should be maintained below a value of 2 to minimise this spurious transport (Griffies and Treguier, 2013;Ilicak et al., 2012). As $U$ increases with the inclusion of tides and other high frequency processes, then maintaining $Re_\Delta$ below this limit becomes more problematic. Beyond this simple criteria, quantifying spurious numerical mixing remains challenging, with a number of different methods having been proposed, each with different assumptions and applicability (Griffies and Treguier, 2013 and references therein; Burchard, 2012; Klingbeil et al, 2014).

When changes to the underlying numerics or refining the grid to at least resolve the barotropic and topographic scales is not practical (e.g. for an Earth System Model), or if numerical mixing remains an issue, making direct tidal modelling undesirable, then the alternative is to use tidal mixing parameterisations, which can be adjusted not to overmix in the deep ocean. These make use of the increasingly fine resolution tidal information available from altimetry constrained models, e.g. TBX08 at 1/30º (Egbert and Erofeeva, 2002). The parameterisations should include benthic and under-ice mixing (Luneva et al., 2015), and mixing by baroclinic tides (St. Laurent et al., 2002). Simmons et al (2004) consider the application of an internal tide energy flux parameterisation, driven by a barotropic tidal model (St. Laurent et al., 2002;Jayne and St. Laurent, 2001), and how to translate this to an interior vertical diffusivity for implementation in a coarse resolution ocean circulation model. In contrast, Allen et al (2010) explore using a 1D mixing model (GOTM) driven at each horizontal grid cell by imposed sea-surface slopes to estimate the vertical profiles of tidal shear. This has the advantage that it can accurately account for the interaction of tidal boundary layers and stratification, which is seen to be important, for example, in the Arctic (Luneva et al., 2015), but does not account for internal tide mixing. Transport by tidal rectification is less easy to parameterise, but is expect to be secondary to the mixing effects on a global scale.

### 3.1.2   Vertical coordinates

Vertical coordinates are a key consideration when modelling the coastal-ocean; the bathymetry necessarily varies substantially at the transition from open-ocean to shelf sea and from coastal seas to the land. As noted above, mixing processes require the accurate resolution of the benthic boundary layer, as do downslope flows such as cascades and Ekman drains. Moreover, bottom boundary mixing and freshwater input lead to exceptionally sharp pycnoclines. For example, an analysis of CMIP5 models by Heuzé et al. (2013) showed that those (few) models that correctly produced Antarctic Bottom Water on the shelves were unable to cascade this water down-slope to the deeper ocean.

This need to increase resolution in shoaling water, alongside the need for smoothly represented across-isobath flows has led to a prevalence s-coordinates in coastal-ocean models, accepting some exceptions (Maraldi et al., 2012;Daewel and Schrum, 2013) that have used z-coordinates. The large majority of global ocean models use z- or isopycnal coordinates. The reasons behind the lack of global s-coordinate models are the well documented issues of calculating horizontal pressure gradient and diffusion terms on sloping coordinate surfaces.

The requirement for tidal simulations to employ a non-linear free surface and sophisticated vertical grid leads to time-varying vertical coordinates with large slopes. This requires the use of complex schemes to derive the horizontal pressure gradient term in order to avoid spurious currents at steep topography (e.g. Shchepetkin and McWilliams, 2003) and an unrealistically energetic inverse energy cascade (e.g. Holt and James, 2006). As with numerical diffusion, accurately diagnosing this issue in realistic model simulations is problematic, so recourse is usually made to theoretical constrains such as the hydrostatic consistency condition to define limits on the steepness of coordinate surfaces. Substantial progress has been made in addressing

this issue through advanced numerics (e.g. Shchepetkin and McWilliams, 2003) or hybrid coordinate approaches (e.g. Siddorn and Furner, 2013); bathymetric smoothing is the last resort, but is still required in some cases. Given the principle of minimising the changes to the model in the open-ocean, the natural choice (for a z-coordinate model) is to move to a hybrid system with z-coordinates in waters greater than a certain depth, transitioning to terrain-following coordinates in shallower water (Shapiro et al., 2013;Luneva et al., 2015;Zhang and Baptista, 2008). These can be formulated to match the original model's vertical coordinate system at the transition depth. Such an approach does require the use of a sophisticated horizontal pressure gradient calculation, but minimises the effect of any residual error from this term in the low dissipative open-ocean region where it is likely to be most harmful (e.g. in feeding spurious energy into the inverse energy cascade). This approach has potentially substantial benefits for mixing and downslope flows. For example, Wobus et al (2013) have shown some success with a mixed z-s coordinate model to facilitate the cascading downslope near Svalbard. It could also be used to facilitate accurate cross-basin transports at deep sills.

The issues with using s-coordinates in climate models are described by Lemarié et al (2012). These include spurious mixing through diffusion associated with the advection scheme. This is particularly problematic as it occurs on steep slopes where physical mixing from (e.g.) internal tides may be prevalent. A solution to this is to use a non-diffusive advection scheme coupled with a rotated biharmonic diffusion scheme (Marchesiello et al., 2009). Another issue is the need for vertical mixing schemes that can accommodate wide variations in layer thickness and still retain low mixing in the ocean interior.

### 3.1.3    Vertical and horizontal mixing parameterisations

Surface mixing processes of wind stress, convection and wave effects are common to open and coastal-oceans, and so the primary consideration for vertical mixing schemes in the coastal-ocean that differ from the open ocean is the need to accurately model mixing at the benthic boundary layer.  Two equation turbulence models (e.g. k-ε) readily accommodate this and by using the Generic Length Scale approach (Umlauf and Burchard, 2003) these can be flexibly incorporated in a global model. While these approaches give good results in shelf seas (Holt and Umlauf, 2008), they differ substantially from schemes used in global models (e.g. the TKE scheme in NEMO and KPP scheme in MOM5). The implications for global ocean simulations, e.g. deep water mass preservation properties and maintenance of the meridional over turning circulation, have yet to be established. Particularly the issue of the k-ε model's performance at low vertical resolution, needs to be established. With the length scale limiter that is usually used with this model, it reduces to a background value inversely proportional to the buoyancy frequency in strongly stratified, weakly turbulent regimes (Holt and Umlauf, 2008; eqn 6 therein). This is broadly consistent with the behaviour of ocean interior internal wave mixing (e.g. Gargett, 1984), so might be expected to give good results with careful parameter selection. This issue has been explored for the KPP model with terrain following coordinates by Lemarié et al (2012) and modifications proposed in the context of these coordinates.

Quasi-horizontal mixing approaches suitable for both open and coastal-ocean require careful consideration. These schemes play two distinct roles: first to represent the effect of unresolved eddies in transport and second to complete the cascade of energy to unresolved scales. The former is particularly important in non-eddying open ocean models (Gent and McWilliams, 1990), but is not generally required in coastal-ocean models. The latter is common across both types of model, and is often treated as a stabilisation term without reference to specific physical principles. Both shelf and global ocean models tend to employ a combination of Laplacian and/or bi-Laplacian mixing for momentum and tracers. Mixing of temperature and salinity usually takes place along isoneutral, rather than geopotential, surfaces and this requires careful implementation in the case of sloping vertical coordinates and at fronts where isopycnals intersect the sea surface and bed. The sloping coordinate systems requires the use of rotation operators for lateral tracer mixing and this can prove less accurate or more challenging for time-varying and highly sloping coordinates (compared to z-level models since isopycnal slopes tend to be fairly close to horizontal), owing to the small-slope approximation (Beckers et al., 2000). This issue has received significantly less attention than similar

considerations with regard to the horizontal pressure gradient calculation, but is explored in this context by Lemarié et al (2012).

As we see above global models span a wide range of dynamic scales and this is exacerbated when shelf seas are considered in detail; a quasi-uniform resolution model generally includes both eddying and non-eddying regions. Hence any model that aims to accurately cross these scales needs to account for the qualitatively and quantitatively changing nature of sub-grid-scale processes. This requires scale-selective approaches to determining sub-grid-scale diffusivities and viscosities (or other forms of closure). The simplest are just depth dependence (Wakelin et al., 2009) or based on horizontal shear (Smagorinsky, 1963). Combination of these with water column density structure (Hallberg, 2013) are likely to be most appropriate, but have not yet been tested in both the open and coastal-ocean contexts.

### 3.1.4    Coastal boundary conditions and rivers

A key feature of the coastal-ocean that needs to be considered is coastlines and related bathymetry (e.g. restricting exchange between regional basins). The treatment of the coastal topology is very much dependent on the horizontal gridding approach. Quadrilateral meshes approximate coastlines by a blocked mask and the resulting representation of the coast is highly resolution-dependent and leads to two specific issues. First the detailed representation of coastal features, e.g. at an inlet or a strait, is limited by this resolution; there is some limited scope to alleviate this through mesh distortion. Second, the staircase representation of a straight coastline impacts the fundamental numerical properties of the model, notably the propagation of Kelvin waves is retarded (Greenberg et al., 2007) and the accuracy of solution is reduced (e.g. from second to first order; Griffiths, 2013), even for coasts very closely aligned with the mesh, with only an occasional step. This can be seen as a special cases of the stepped representation of topography by z-coordinates models; the issue of bottom topography representation is alleviated by using terrain following coordinates. Available solutions at the coast-line for quadrilateral meshes are through shaved cell (Adcroft et al., 1997;Ingram et al., 2003) or immersed boundary (Tseng and Ferziger, 2003) approaches for high resolution models, or porous barriers (Adcroft, 2013) for coarser resolution models. Triangular mesh models, when paired with terrain following coordinates, do not encounter these issues: they can fit the coastline with an arbitrary degree of accuracy limited by the minimum acceptable scale and accuracy of the geographic information. The representation of the details of the coastline is key advantage of triangular mesh models. However, even with highest resolution models being considered the accurate details of coastlines will not be well represented, particularly in bays, estuaries, fjords, etc and these must be left to local scale models (often of a few 100m's resolution), include more detailed processes such as the capability to wet and dry with the tide. Similarly for the resolutions considered here, parameterisation of riverine effects are still required for an accurate representation of their transport processes. This can be achieved by, for example box modelling approaches, as currently being tested in the Community Earth System Model (Bryan et al., 2015).

### 3.2    An example of improved resolution and process representation in comparison with CMIP5 models

To illustrate what might be achieved through higher resolution, introducing tides and sophisticated mixing schemes in the context of the coastal-ocean we consider runs of the global ORCA12 (Marzocchi et al., 2015;Duchez et al., 2014) and the Northern North Atlantic (NNA; Holt et al., 2014) NEMO models. Both use the same NEMO code base (V3.5) and have 75 z-partial step layers in the vertical. The ORCA12 model uses a TKE vertical mixing scheme, a filtered free surface formulation and does not include tides. The NNA model is an extraction of the grid and bathymetry from ORCA12 that includes tides, a k-ε vertical mixing scheme (implemented by the GLS approach; Umlauf and Burchard, 2003), with log-layer bottom friction and a mode-split explicit free surface with variable volume. Both use DFS surface forcing (Brodeau et al., 2010) and NNA takes lateral boundary conditions from ORCA12. For brevity, here we focus on the Potential Energy Anomaly (PEA; Eqn 2) as a measure of upper ocean stratification and Figure 4 shows both these models perform substantially better than the CMIP5 models in both RMSE and correlation across the whole domain. The same is true for the correlation in the coastal-ocean, but

there are two 0.5° CMIP5 models (NorESM-ME and CNRM-CM5) of comparable performance in terms of RMSE. No significant difference between NNA and ORCA12 is apparent in these overall statistics.

Figure 5 shows the mean July PEA for these two models to examine the differences in stratification in more detail. It shows that while the differences across the whole region are comparatively minor, the differences in some localised regions are very marked. A particular example is in the southern North Sea, English Channel and Irish Sea, where the expected well mixed regions are much clear in NNA than ORCA12, and more in accord with observations (Holt and Umlauf, 2008).

To explore how well these models reproduce the seasonal cycle in stratification, the mean annual cycles of PEA for 10 regions (see Figure 5) are shown in Figure 6; this is limited to water depth <500m to focus the comparison on the coastal ocean. Each region is selected to cover sufficient data to give a reasonably smooth mean seasonal cycle in the observations, but the results are inevitably dependent on the details of this choice. Moreover, because EN4 is not a systematic data set, deriving a mean annual cycle in this way potentially mixes inter-annual and spatial variability (for further discussion on this see Holt et al., 2012). Nonetheless, it provides a useful guide to model performance. Alongside NNA and ORCA12, results are shown for the same 19 CMIP5 model as in Figure 4 and the overall RMSE, the median and minimum RMSE values for CMIP5 and which model is the lowest. We see that NNA has a lower RMSE than ORCA12 in all regions except 1 and 9 (Norwegian Sea and Georges Bank, where both models exhibit no significant improvement on CMIP5) and 4 (southern North Sea, where both have small errors). This demonstrates a clear advantage of this combination of process representation (i.e. including tides and a two-equation turbulence closure scheme). Apart from in regions 1 and 9, the NNA and ORCA12 models improve on the median error from the CMIP5 models; this is not a remarkable results and it would be worrying if it were not the case that high resolution, reanalysis forced models could not out-perform coarser resolution coupled models. However, what is more interesting is that in most regions the best of these CMIP5 models out-performs or is very close to either NNA or ORCA12. This may well happen by chance: there is a broad spread of CMIP5 models results here and 6 different models are the highest performers. However, some of the higher resolution models (CNRM-CM5, 0.5°; MPI-ESM-MR, 0.3°; NorESM1-ME, 0.45°) lead (among this CMIP5 ensemble) in 6 out of 10 regions, which deserves further investigation. For example the CNRM-CM5 (Voldoire et al., 2013) and NorESM1-ME (Bentsen et al., 2013) both include the tidal mixing parameterisation of Simmons et al (2004).

Some other aspects are clear from this comparison. These CMIP5 models generally over-estimate the PEA and its annual cycle. This is particularly apparent in the eastern boundary up-welling regions (5 and 6) where the observed annual cycle is very small. Such biases are not apparent in NNA or ORCA12.

While a much more comprehensive assessment is require to inform the appropriate aspects of model development, this does demonstrate some clear advantages to improved resolution and process representation. It also identifies some areas for further investigation, notably the biases on the eastern U.S. coast.

### 3.3 Resolving the pertinent scales

The most significant challenge in representing the coastal-ocean in global models relates to the small scales needed to represent the processes and geography (coastline, bathymetry, straits) of these seas. There are essentially two options for achieving a refined horizontal resolution: either increase the quasi-uniform resolution of the whole grid or introduce a multiscale capability that allows refinement in specific locations. We consider briefly what these capabilities might be below, but first explore the balance between these two options if we desire to resolve a particular set of processes globally, refining the model locally to achieve this. We quantify this conceptually, with no consideration of mesh structure, by building on the scale analysis above and define:

$$\underline{N}_x = \sum F^2 = \sum (n/e)^2 = \sum (n \max(\Delta X, \Delta Y) . E/L_x)^2 \tag{3}$$

to define the global sum of the number of cells needed in each global model grid cell to resolve a process, characterised by length scale $L_x$ at a particular level ($n$). Following the discussion in section 2.1 we take $n$=10, 3, 2 for $L_{bt}$, $L_T$ and $L_1$ respectively. A constraint is imposed on this:

$L_{min} < L_x / n < \max(\Delta X, \Delta Y).E$                                                                                 (4)

The upper limit specifies a 'base' resolution, i.e. a multiple ($E$) of the global ORCA12 grid (resolution: $\Delta X, \Delta Y$) that is being refined. The lower limit, $L_{min}$, acknowledges that there are limits to how fine a resolution is desirable, particularly in the case of scales that tend to zero with the water depth, and with respect to timestep constraints.

As an example, Figure 6 shows how a $1/12°$ ORCA tripolar grid might be refined to a minimum scale of 1.5km (~$1/72°$) as required by the above criterion, with $L_x$ being the smaller of the baroclinic, barotropic, and topographic scales. Values at each cell range from $F^2=(n/e)^2=1$ (no refinement) to $(\max(\Delta X, \Delta Y).E/L_{min})^2$, =36 in this case. Mid-latitude and arctic shelves require modest refinement ($\times$10-15 extra cells); the reduced based mesh size of the ORCA grid counters the reduced Rossby radius here (noting the absolute values of $F$ are dependent on this grid structure). In some very shallow tropical regions the number

is at or close to the maximum value, indicating that the desired level of process resolution is not always achieved. The accuracy of this estimate is limited by the underlying information (notably the bathymetry and the Rossby radius) and no consideration of the refinement needed to resolve the coastline is made. Nonetheless this still provides a useful guide in terms of the relative cost of multiscale and globally-refined resolution approaches. Here we compare this calculation with the total number of grid cells in the globally-refined case (at $L_{min}$). Because the minimum scale is the same for both no timescale factor is needed. This

approach takes no account of the mesh structure needed. In particular there will be limits on how quickly scales can be allowed to vary on an unstructured mesh (see for example Figure 6 lower panels, show the change in resolution needed can be locally very abrupt) and so this puts a lower bound on the number of cells needed in the multiscale case.

We consider three values of $L_{min}$: 9.3km (~$1/12°$), 3.5km (~$1/36°$) and 1.5km (~$1/72°$) (c.f. Table 2) in Figure 8. So for example, a $1/4°$ global model refined where necessary to resolve the smallest of these scales down to a minimum scale of 9.3 km (left

pannel) requires about 0.25 the number of grid-points of a full $1/12°$ grid, or a saving of about a factor of 4. As the minimum scale decreases to 3.1km (middle panel) and 1.5 km (right panel), the saving increases to 0.095 (10 times fewer points) and 0.046 (factor 22), compared to the full global grid at the minimum resolution. Similarly, a $1/12°$ model refined to a minimum scale of 1.5km has 0.06 (factor 17) times fewer cells than a $1/72°$ global model. The limiting behaviour evident from these plots arises because at coarse base resolution most of the grid is refined to meet the criteria (i.e. the base resolution becomes

irrelevant), while at a fine base resolution this meets the criteria in many regions anyway and the refinement becomes less relevant. These results are considered in terms of what may be computationally practical in section 4.

### 3.4 Options for multiscale modelling

There is already a substantial literature on multiscale modelling and we do not attempt to review this here. Unstructured mesh approaches generally focus on triangular mesh models using a finite volume approach e.g. FVCOM (Chen et al., 2003);

FESOM2 (Danilov et al., 2016) or a finite element approach, e.g. FESOM1.4 (Wang et al., 2014), SELFE (Zhang and Baptista, 2008) and SCHISM (Zhang et al., 2016). In contrast MPAS (Ringler et al., 2013) is based on hexagonal meshes using a finite volume approach. Danilov (2013) provides an account of the issues of unstructured mesh modelling, and what is clear from that review is that selecting an solution approach or grid arrangement, for example, on the basis of a lack of computational modes or formal accuracy is far from straightforward, and must be left to detailed investigations in idealised and realistic cases.

Structured grid models have scope for multiscale capability by distorting their horizontal coordinates and through nesting. Coordinate transformations generally limit the refinement to a single region of interest. An example to facilitate regional impact

studies is the use of a rotated polar grid (Gröger et al., 2012) to focus resolution on European seas. While this can address the downscaling issue for a single region, it does not help with the upscaling question.

Nesting is the most common approach to multiscale modelling. In its simplest form boundary conditions for a fine resolution regional model are taken from a previous run of a larger area ocean model. It has the significant advantage that the global model does not have to be rerun for each regional simulation. There is, however, the practical consideration of the effort required to setup and test a new regional configuration for each new area of interest. Nesting remains an important approach for investigations of regional systems, and providing fine scale information, e.g. for operational or research purposes. The general downside to nesting is the accuracy at which information can be exchanged between the two domains and the degradation of the solution at the boundary; it is usual to linearise the boundary conditions and to only exchange a limited subset of information at lower frequency than the model timestep. That said, there has been extensive work on regional model boundary conditions (e.g. Marsaleix et al., 2006;Mason et al., 2010) and by using a careful combination of active and passive approaches good solutions can be obtained. One-way nesting can be straightforwardly extended to a global scale using multiple regional nests (Holt et al., 2009). The problem is simply one of standardising the configuration procedure and of managing workflow. However, one of the key advantages of regional models, that they can be tailored to the specific conditions of a region, is generally lost in automatically configured domains. The underlying assumption to such one-way nesting is that feedbacks between the regional and global simulations are small, at least on the timescales of interest and again it only addresses the downscaling question.

A natural extension of the nesting approach, which allows for upscaling, is two-way global scale nesting. The AGRIF tool (Debreu et al., 2012) provides a capability to automatically generate nests, which has been utilised in both the ROMS and NEMO systems (e.g. Biastoch et al., 2008), generally with individual regions being refined with one or more nests. In theory this is extendable to the global scale, with multiple nests placed to locally resolve coastal-ocean processes. Several approaches exist to couple the two grids, reviewed by Debreu and Blayo (2008). Because this occurs 'in memory', these can be substantially more sophisticated than off-line nesting by file exchange, and essentially aim to link solution approaches in the two grids, coupling at the time steps of the respective grid. This means that as well as having two-way interaction, many of the issues associated with off-line boundary conditions noted above are alleviated, although noise and wave reflection are two issues that require particular attention. An issue with this approach in the global context is the restriction (for AGRIF) to rectangular domains (in model coordinate space; see below). This is somewhat inefficient and inflexible, and the coupling between neighbouring refined regions, with potentially different levels of refinement needs to be considered. Large irregularly shaped nests (e.g. Holt et al., 2009) would be good option, not over-refining in the open ocean and limiting the number of grids and connections between them This would require substantial development to AGRIF or an alternative approach. An approach that has yet to be thoroughly explored is using model couplers (e.g. OASIS3-MCT) as a 2-way downscaling tool. This would allow complete flexibility between nests, e.g. a different executable can be run in each nest, but whether the coupling system is sufficiently efficient to permit coupling at the model time-steps is unclear.

A key limitation to multiscale models is time stepping, which is closely related to the scalability of the models, discussed below. The trade-off is between explicit models, which are computationally efficient with MPI parallelism, but have a time step limited by the CFL condition and implicit models, which are not limited by the CFL condition, but are less computationally efficient with MPI parallelism, due to the need for global matrix inversions. Currently the balance is towards explicit time-stepping models (e.g. with time splitting between barotropic and baroclinic modes), given the use of global models on many thousands of processor cores. This has implications for multiscale approaches: the approach must either accept the limitation of the smallest scale in the grid, introduce some level of implicit time stepping, with consequent implications for scalability, or else introduce a locally refined time-stepping approach, whereby different timesteps are used in different regions. The latter is natural for the multi-blocking approach (and is assumed in the analysis below), but is highly complex for unstructured mesh

multiscale approaches (e.g. Dawson et al., 2013). A move to implicit timestepping, aside from any accuracy/diffusion issues, requires the development/use of very efficient numerical solvers.

To put global nesting in the same context as the above scale analysis, we consider a multi-block approach (accepting the limitation to rectangular domains for now), and consider the global ORCA12 grid divided into ~15$^{o}$x15$^{o}$ blocks. Each of these is then given a refinement level $F^2$ ranging from 1 to 36, as above. To provide a representative maximum value (but not set by a very few large grid point values), this is taken to be the 95$^{th}$ percentile of the grid cells in each block. To mimic the AGRIF refinement process each block takes an integer value: $(int(F))^2$. This example leads to 194 out of 344 cells requiring refinement (Figure 9). Such a setup would be a challenging computational engineering effort and certainly less elegant than an unstructured mesh approach, but maybe be more efficient (quantified below) and is available as an evolution of the structured mesh approach, common to most of the current climate scale global modelling effort (and so building on the expertise therein), rather than a move to a radically different approach. Whether it is more or less accurate than a comparable finite volume or element unstructured mesh approach must be left for future investigation.

## 4    Utilising the computational resources

### 4.3  Trends in High Performance Computing

Ocean modelling has benefited from the general exponential growth in High Performance Computing (HPC) capability, with the largest machines approximately doubling in performance every 18 months since 1993 (TOP500[1] list). There are two technology drivers for this: firstly increases in clock speed and improvements in architecture (particularly Instruction Level Parallelism) and secondly massive increases in parallelism. In 1993 the TOP500 list still contained machines with only one processor, in June 2016 the smallest system had 5310 cores and the largest had over 10 million cores. The first of these drivers has largely stalled as clock speeds have peaked at around 2-3 GHz due to power density limitations. Instruction level parallelism has also peaked at around 4-8 instructions per clock cycle; memories are not fast enough to provide enough operands to justify greater values. Further performance increase into the future is therefore expected to be driven solely by an increase in parallelism, through larger and larger number of processor cores.

Continuing the current exponential growth towards exaflop performance (10$^{18}$ operations per second) specifically requires a substantial reduction in power consumption (by ~100-fold) to keep the power costs of HPC systems within reasonable limits. If these power efficiency constraints are lifted to achieve exascale systems there are two major impacts for ocean modelling. First is the prospect of a single ocean model running at exascale performance levels on e.g. 100 million cores. Alongside this there would be a knock-on impact of smaller systems as petascale systems become available with about 100,000 cores in a single rack, consuming only ~100kW, and so accessible by the modelling community at an institutional level.

To use the UK research community perspective as a practical example, Figure 10 shows the increase in the peak performance of the UK Research Councils' (RCUK) HPC facility, from HPCx in 2006, through the four phases of Hector to the current machine Archer[2]. The peak performance of this facility has increased exponentially over the past ~10 years, although the general trend has flattened off since the rapid increase between HPCx and Hector Phase 2a. A conservative estimate is to extrapolate the trend from Phase 2a to Archer Phase 2. This gives a peak performance of ~13 times Archer Phase 2 by 2019 (32Pflop/s) and ~745 times by 2023 (610Pflop/s). This closely follows the TOP500 trends, and predicts the UK maintains a performance about a factor of ten lower than the US at any one time (or lags by 3-4 years). There are of course many unknowns in this projection such as the size of the overall research community and share of the resource which the marine science sector may receive. Nonetheless, this usefully quantifies the often quoted remarks around continually increasing computer power and puts bounds on what may be expected.

---

[1] http://www.top500.org/

[2] www.archer.ac.uk

In terms of ocean model design, to effectively utilise large numbers of cores, codes will have to extract very high degrees of parallelism from the underlying numerical algorithms. This requires at least three-way nested parallelism with high-level coarse-grained parallelism at the node level probably using MPI, multi-threading on a node using OpenMP or OpenAcc, and fine-grained parallelism within a core, e.g. vectorisation at the loop level. Memory management will become increasingly important. The size of memory cannot increase to match the numbers of cores, on ground of cost and power, and the amount of memory per core is expected to reduce significantly (although memory per core is still relatively stable in the example presented here; Figure 10). Memory bandwidth per core and interconnect speed per core is also expected to drop. Algorithm design must therefore focus on management and movement of data in memory and between nodes.

### 4.4 Scalability and efficiency of ocean models

An important distinction needs to be made between computational resource (CPU hours) and model simulation time (Simulated Years per Day; SYPD). The available computer resource is generally increasing through increased parallelism (more cores per chips, more CPUs and novel, accelerator-based architectures), and is what is metered and limited by computer centres. It is, however, the SYPD that limits the science that can be done with a particular model (assuming the resource is available). For example, Table 3 lists anecdotal reports of turn-around time (SYPD) for three global ocean models (MOM6 GFDL CM2.6; NEMO ORCA12 and FESOMV2 Glob15) and a high resolution coastal-ocean model (NEMO AMM60), also shown is this valued scaled with the grid cells (horizontal and vertical) per core and the time step to give a rough comparison of the efficiency. The two structured grid global models have a comparable efficiency (2.35 and 3.5 SYPD; 207 and 223 kTimestep gridcells/sec; kTGPS), whereas the unstructured mesh model is somewhat more efficient (17 SYPD and 396 kTGPS; discussed further below), but is run on considerably fewer processors.

Currently the minimum efficient size of model run by each MPI process is about 20x20 or 400 grid cells per core. As the resolution reduces, then, for an explicit time-stepping model the CFL stability criteria requires the time step to reduce and the model runs more slowly (reduced SYPD), irrespective of the increased resource. For example, the $1/60^\circ$ resolution NEMO model of NW European continental shelf achieves only ~1 SYPD, but is somewhat more efficient at 245 kTGPS than the global NEMO model. For large scale climate and earth system model simulations, with substantial resource available, but a requirement to complete many centuries of simulation in a restricted time period, this is the key limitation. When running large numbers of shorter simulations in research mode (e.g. by a whole research community), the resource (CPU hours) itself provides the limit. Given static CPU speeds there are two options to mitigate this reduction in SYPD: modify the model numerics (with respect to time-stepping; see above) and/or improve the parallel scalability. The latter can occur in two ways. Firstly, by reducing the size of the sub-domain within an MPI process that can be used efficiently, essentially by reducing the ratio of communication costs to computation costs, e.g. by using larger halos to increase message size and reduce latency effects. Secondly, by introducing alternative levels of parallelism, so that more cores can be used efficiently by a single MPI process, e.g. using multi-threading (loop-level parallelism) with OpenMP or OpenACC. Another possibility which is being explored is parallelization of the time-domain, especially attractive for long time duration low-resolution climate runs. Parallel-in-time methods offer the prospect of another two orders of magnitude in concurrency (Haut and Wingate, 2014), but are still at the early research stage.

An important and complex question is whether unstructured mesh models are inherently more computationally expensive than structured grid ones. They are certainly more complex, with more floating point operations required per degree of freedom and also require indirect memory addressing (only in the horizontal, assuming a structured vertical data structure). However, we are moving to a computational situation where 'flops' form a minor part of the cost, and the cost of indirect addressing can be effectively 'hidden' by sufficient vertical computation. Triangular meshes require more edges to span a particular domain with comparable resolution to quadrilaterals, but equally can have fewer edges if areas can be identified where reduced resolution is required. They have an advantage over structured mesh models in that computation is only over 'sea'-points, but this is a

marginal advantage with a large number of cores when land-only cores are excluded and/or sophisticated load-balancing algorithms for domain decomposition are used (e.g. k-partioning; Ashworth et al., 2004). There is no *a-priori* reason why either class of model is more or less scalable than the other, simply on the basis of grid and data structure: the answer in practice lies in the details of numerics, e.g. choice between implicit and explicit time-stepping and the number of halo-exchanges

needed for high-order advection schemes, and the success of the optimisation in a specific context. The higher computation per degree of freedom may weigh in favour of unstructured grid models in terms of relative scalability. Historically, there has been a substantially computational penalty e.g. the finite element model FESOM1.4 was reported to be ~10 times slower than comparable structured grid models (Wang et al., 2014)  and experience with the FVCOM in the NW European shelf suggest this model is ~5 times slow than NEMO. More recent work suggests a very different picture: MPAS quotes a penalty of 3.4

compared with POP (Ringler et al., 2013) and the finite volume FESOM2 (Danilov et al., 2016) code is reports a through-put 5 time faster than FESOM1.4. The comparison of this model with two finer resolution structured grid models in Table 3 suggests this model is, if anything more efficient. This may be because the time step ratio (of 3) between this model and the NEMO and MOM6 cases is substantially larger than the ratio of nominal grid resolution resolutions (15/km9.3km = 1.6; c.f. the ratio of efficiencies in Table 3: 1.6); i.e. the unstructured mesh model achieves a longer time step maybe because it can

have a more uniform grid (recalling the finest cell in the ORCA12 grid is 1.3km). So these anecdotal results (accepting different problem sizes, runs with different processor counts, on different computers are being compared here) suggests parity in resource cost and turn-around time between present day structured and unstructured mesh models is a realistic prospect. However, the structured grid models are themselves being continually and extensively optimised (e.g. for openMP parallelisation) so there is also the possible that a gap similar to the MPAS-POP comparison persists.

**4.5  Exploiting Future HPC Architectures**

Exploiting Petascale or Exascale levels of performance will require substantial algorithmic development to achieve the required level of concurrency. Many researchers have been looking at ways to improve the parallel scalability of ocean models on massively parallel architectures. Within the context of NEMO there has been work looking at hybrid MPI/OpenMP parallelisation strategies (Epicoco et al., 2016) and a port of the code using OpenACC directives targeting accelerator-based

architectures (Milakov et al., 2013). The layered approach to software design (Ford et al., In Press) provides one way to achieve this, while retaining code that can still be straightforwardly developed by an ocean modeller. The key idea in this approach is the PSyKAl (Parallel System, Kernel, and Algorithm) Separation of Concerns. The ocean modeller should not have to be concerned about the (ever-increasing) complexity of the underlying computer and the computational scientist who optimises the code should not have to understand ocean processes. In practice, the separation is achieved by using domain-specific

knowledge about the type of problem being solved; in particular, the fact that the majority of the available parallelism comes about through performing the same computations at each point in the model mesh. In PSyKAl, the ocean modeller is responsible for writing the Algorithm and Kernel layers while all performance optimisation (including all code related to parallelism) is restricted to the PSy layer. The Algorithm describes the model computation in terms of logically-global fields (e.g. add field1 to field2) while Kernels implement the actual computation to be performed at a grid point, and the PSy layer

distributes this across the model domain with a particular parallel optimisation approach.

This approach is in use for the new, finite-element LFRic atmosphere model being developed by the UK Met Office. It has also been applied to two different, finite-difference shallow water models. The first, 'shallow', is a benchmark code originally developed by Paul Swarztrauber of NCAR. The second consists of (only) the free-surface component of NEMO and is therefore named 'NEMOLite2D'. It has been demonstrated that any loss in performance resulting from the PSyKAl re-structuring of

these codes can be regained by optimising the PSy layer (Porter et al., 2016). The middle, PSy, layer of such models can be automatically generated from a knowledge of the Kernels (as described in meta-data) and the Algorithm. Since this generation can be tuned to match a particular computer architecture (e.g. CPU or GPU) as well as being used to support different forms

of parallel execution (e.g. distributed versus shared memory), the aim is that the model as a whole will be performance portable while requiring no changes to the natural-science parts of the code. There are also substantial software engineering benefits to this approach in that scientists no longer have to worry about writing correct parallel code (e.g. the "Do I need to do a halo-swap?" problem) and optimisation experts are protected from introducing errors in the science.

## 4.6 The comparative cost of ocean models

To link the scale analysis and the computational issues, Table 4 lists a set of possible future model configurations, and an estimate of their relative resource cost with and without a timestep penalty. The relative resource cost is calculated by:

$$C = \frac{N}{N_{25}} \frac{L_{min25}}{L_{min}} S, \tag{5}$$

where $N$ is the total number of grid cells in a configuration defined by $L_{min}$, and $N_{25}$ and $L_{min25}$ are the reference values for nominal $1/4^o$ global NEMO configuration, ORCA025. The inverse ratio of lengths scales is included in Eqn 5 to introduce a global time-stepping penalty on the assumption that local time-stepping approaches have not been implemented (except in the block-refined approach). $S$ is a factor for models that need unstructured meshes, which following the discussion above we take to range from 3.4 (e.g. Ringler et al., 2013) to 1 (parity between structured and unstructured cases). Here we focus on resource rather than turn-around time (SYPD) and, without the scalability and timestepping improvements identified above, an increased resource may only be utilisable by waiting longer for a finer resolution model to finish. Moreover, this simplistic cost-model ignores all the real-world issues that would have to be faced, notably the changing balance between computation, memory access and communication, and also all arising data handling and storage issues.

Three quasi-uniform structured meshes, three unstructured mesh multiscale options and an example of a block-refined multiscale case are considered. For the block-refined approach, we assume a variable timestep and assign a step penalty (reducing it by 1/F) for each block independently. This assumes the model is load-balanced and optimised for the finest meshes, so again the measure here is resource used rather than time to completion. To estimate when these could become routine models, an exponential fit to the growth of RCUK computer peak performance (Figure 10) is used so:

$$Y = int(\log_{10}(C)/P + Y_0), \tag{6}$$

taking the $1/4^o$ model in 2011 as a base line ($Y_0$) for a 'routine' high performance global physical oceanography research model. From Figure 10, $P=0.258$ yr$^{-1}$ (i.e. doubling every ~1.2 years). So, for example in 2017 a $1/12^o$ model uses a comparable fraction of the total computer resource available as a $1/4^o$ model in 2011. There are many caveats to these estimates, not least the scientific development time needed to achieve the various stages, but they do serve as a reasonable guide to either encourage or constrain aspirations.

A key milestone in this growth is a $1/12^o$ global model refined to $1/72^o$ to resolve coastal-ocean processes. This represents the amalgamation of the current state of the art of global and regional scale coastal-ocean modelling. When this would be comparable to a $1/4^o$ model in 2011 for an unstructured mesh multiscale approach depends critically on the efficiency in the unstructured modelling technology. If these achieve parity with present day structured grid models ($S=1$) then this point is reach at 2021, if factors similar to the present day MPAS experience persists then this date becomes 2023. The estimate for the block refined multiscale approach is 2022. All of these are sufficiently ahead of the figure of 2026 for a $1/72^o$ global model, assuming static computational efficiency for the structured grid model (i.e. the development effort is primarily toward scalability and reducing SYPD rather than reducing resource requirement). This sets a clear challenge for ocean model developers and computer scientists to develop an efficient and accurate multiscale approach by this date.

The considerations above have focused on high resolution physical ocean models, e.g. as part of a coupled climate model or an operational forecast system. For Earth System Models with complex marine and land surface ecosystem and atmospheric chemistry components, we must accept that the 'routine' model of today (2016) is a $1^o$ resolution ocean. The scaling then

suggests that a 1/12º global and a 1/12º global model refined to 1/72º would not have a comparable computational cost to a nominal 1º ocean model until 2027 and 2035 respectively. This suggests options to improve the coastal-ocean in centennial scale ESM simulations (e.g. for fully coupled carbon cycle simulations) will remain highly parameterised for at least the next decade, and for fine scale processes, two decades.

## 5. Conclusions

The analysis and investigation presented here suggest the prospects for improving the representation of the coastal-ocean in global models are now promising. We can identify three concurrent avenues of development to achieve this. Firstly, global models are now routinely run at the horizontal resolution of past shelf sea model simulations that capture many of the pertinent scales, and with dynamics that allow the representation of relevant processes, such as split-explicit time stepping rather than long wave-filtered or implicit approaches. In this case some (comparatively) straightforward developments can be included in the simulations to significantly improve the representation of the coastal-ocean. These are: i) including tides, their generating forces, self-attraction and loading and wave drag effects; ii) using vertical coordinate systems that retain resolution in shallow water, resolve the benthic boundary and allow smooth flow over steep topography; iii) adopting vertical mixing schemes that represent mixing at the surface, pycnocline and benthic boundary layers. These are all existing features of regional ocean models and the general challenge here is ensuring the introduction of these features does not compromise the deep and open ocean simulation, or significantly increase the computational costs; the single example in Table 3 suggests this would not be the case. Further developments to achieve this are likely, for example through non-diffusive advection schemes and quasi-isopycnal vertical coordinates. We quantify the benefit of improved process representation within the context of the current state-of the art in global resolution. This shows substantial benefits in including tides in terms of reproducing the seasonal stratification cycle, although interestingly two of the CMIP5 models (including tidal mixing parameterisations) perform particularly well.

The second area of development is the continued refinement of horizontal resolution to the point that the pertinent scales are well resolved (estimated to be ~1.5km). This is the case in the current generation of region models, and the analysis presented here suggests it would be computationally practical in about a decade's time. The options considered here, in very general terms are: a continued refinement of the quasi-uniform structured mesh, some form of unstructured mesh (presumed to be either finite element or volume), or else a multi-blocking refinement (whereby rectangular regions are refined to a fraction of the parent mesh and two-way coupled to it). The block refinement and unstructured mesh approaches show significant advantages over the refined structured mesh using the objective refinement criteria and very simple cost model considered here. The BL approach using ~13 times less computational resource. The resources needed for the unstructured mesh approach depends critically on the relative performance of this class of model, here we estimate 5-17 times less resource depending on how close to parity with structured grid models the unstructured models achieve.

These results need to be seen alongside the needs of the open-ocean model. For example, Griffies et al (2009) note (in the context of mesoscale eddies): "There is no obvious place where grid resolution is unimportant". The refinement criteria we have considered here, while chosen for coastal ocean processes, have been applied globally. It is apparent that modest resolution refined to the level of current high resolution global models have only marginal benefits when an objective refinement approach is used. For example, a ¼ degree model refined to 1/12º only differs from a 1/12 º global model by a factor of 4 times few grid cells: much of the ocean is refined to meet (e.g.) the Rossby Radius criteria globally. If the criteria was extended to include additional aspects, e.g. ocean variability, (Sein et al., 2016) then this factor will reduce, and the benefits of the multiscale approach become less apparent. If, however, fine resolution process representation is desirable then the scaling clearly favours multiscale modelling, and if we are sufficiently confident in the refinement criteria to use a coarser base

resolution than would be otherwise be chosen (i.e. to allow a degree of coarsening from a contemporary high resolution model) then the multiscale approach can achieve a substantial reduction in the resource needed.

The final area of development, and by no means the least important, is the improved representation of the coastal-oceans through improved process parameterisation. This essentially uses fundamental theoretical and empirical understanding to make up for deficiencies in the dynamical approach and the computational resource. This covers both processes that would not be resolved by any scales considered here and the cases where significant horizontal refinement is not practical (e.g. centennial scale ESM's). Particular areas that deserve attention are: tidal mixing, topography and coastlines, horizontal mixing schemes that account for the large change in scales at the ocean margins, and river plumes. Given that the scale analysis presented here suggests we may be one or two decades away from a well-resolved coastal-ocean routinely run in fully coupled complex ESM's, then these parameterisations are paramount.

This conclusion describes three complementary strands of work, which together have the potential to make substantial progress on our ability to model the coastal-ocean at a global scale, and so our ability to simulate global change and its impact on the societally pressing questions.

## 1. Code and data availability

NEMO model code used to run the Northern North Atlantic Model configuration can be obtained from:

forge.ipsl.jussieu.fr/ipsl/forge/projets/nemo/svn/branches/NERC/dev_r3874_FASTNEt

Data used to prepare Figures 1a-c, 2, 4, 5, 6, 7, 8 is provided at

ftp://ftp.nerc-liv.ac.uk/pub/general/jth/GMD_Holt_GloabalCoasts/

CMIP5 data is available from pcmdi.llnl.gov/search/cmip5/

Data used for Figure 1d from: volkov.oce.orst.edu/tides/TPXO7.2.html, and for Figure 3 from:

www.cgd.ucar.edu/cas/catalog/surface/dai-runoff/coastal-stns-Vol-monthly.Constructed.wateryr-v2-updated-oct2007.nc

Figure 4 and 6 uses EN4.0.2 profile data from www.metoffice.gov.uk/hadobs/en4/download-en4-0-2.html .

Information for Figure 10 was obtained from www.hpcx.ac.uk/services/hardware/, www.hector.ac.uk/service/hardware/ and www.archer.ac.uk/.

## 2. Acknowledgements

This work was supported by the NERC Next Generation Ocean Dynamical Core Roadmap Project, and the National Capability programme in ocean modelling at NOC and PML. Allen is supported by the NERC Integrated Modelling for Shelf Seas Biogeochemistry Programme.  Hewitt and Woods were supported by the Joint UK DECC/Defra Met Office Hadley Centre Climate Programme (GA01101). Harle is supported by the NERC RECICLE project.

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

Table 1: Physical process horizontal scales in coastal and shelf seas

| Process | | Horizontal scale | Reference |
|---|---|---|---|
| Barotropic Tide | $L_{bt}$ | $\sqrt{gH}/\max(f,\omega)$ | (Huthnance, 1995) |
| Tidal excursion | $L_e$ | $U_T/\omega$ | (Polton, 2014) |
| Topographic steered barotropic current | $L_T$ | $H.(\nabla H)^{-1}$ | (Greenberg et al., 2007) |
| Front/frontal jet, coastal upwelling | $L_1$ | $C_{iw}/f$ | (Huthnance, 1995) |
| Baroclinic eddy | $L_E$ | $\pi L_1$ | (Griffiths and Linden, 1982) |
| Internal wave/tide | $L_{iw}$ | $C_{iw}/\omega$ | (Huthnance, 1995) |
| Coastal current/river plume | $L_r$ | $(2Qf/g')^{1/2}(\nabla H)^{-1}$ $(2Qg')^{0.25}/f^{0.75}$ | (Yankovsky and Chapman, 1997) (Avicola and Huq, 2002) |

Here: $U_T$ tidal current, $\omega$ frequency, $H$ water depth, $g$ gravitational acceleration, $f$ Coriolis parameter, $C_{iw}$ internal wave phase speed, Q riverine volume flux.

Table 2 A selection of current model grids

| Nominal resolution | Scale at equator (km) | Global application | Coastal-ocean application | Examples |
|---|---|---|---|---|
| 1º | 111 | Typical of Earth Systems Models CMIP4 and 5 | NA | HadGEM3; (Hewitt et al., 2011), HadCM3; (Gordon et al., 2000) |
| 1/4º | 25 | CMIP6 ESMs | Historical | HadGEM3(Williams et al., 2015) |
| 1/12º | 9.3 | Next generation coupled | Shelf scale/ocean margin | ORCA12; (Marzocchi et al., 2015) AMM7; (O'Dea et al., 2012) AMM12; (Wakelin et al., 2009) |
| 1/36º | 3.5 | Next generation forced | Shelf Scale | IBI (Maraldi et al., 2012) ECOSMO (Daewel and Schrum, 2013) |
| 1/72º | 1.5 | NA | High res. shelf/coastal | HRCS (Holt and Proctor, 2008) AMM60 (Guihou et al In Prep.) |

Table 3 Reported turn-around time of three global models: the MOM6 and NEMO structured grid and the FESOM fine volume triangular mesh model; and a coastal-ocean NEMO model of the Northwest European shelf, AMM60, shown as Sumulated Years Per Day (SYPD). Also shown is an over-all efficiency scaling factor in kTimestep. Gridcells per second (kTGPS).

| Model/ Config. | Nominal resolution | Vertical Levels | Cores for simulation | Grid-cells/core | Time step | SYPD | kTGPS | Reference |
|---|---|---|---|---|---|---|---|---|
| MOM6 CM2.6 | 1/10º | 50 | 10,000 | 972 | 300 | 3.5 | 207 | Dr M. Ward, Australian National University; Per Comms |
| NEMO ORCA12 | 1/12º | 50 | 8972 | 1040 | 300 | 2.3 | 223 | Dr A. Coward, NOC; Per Comms. |
| NEMO AMM60 | 1/60º | 75 | 2000 | 806 | 60 | 1.0 | 245 | Dr J. Polton, NOC; Per Comms |
| FESOM V2 Glob15 | 15km | 46 | 1728 | 1150 | 900 | 17 | 364 | Danilov et al., 2016 |

Table 4: Possible model grids, their costs (Eqn 5) and when they might be computationally equivalent to ORCA025 model (nominal 1/4º) in 2011 based on Eqn 6, from Figure 10. Unstructured grids are refined to resolve the minimum of $L_1$, $L_{bt}$, $L_T$ according to Eqns. 3 and 4. The blocked refined approaches allows timestep to vary between block, other cases it is limited by the global minimum scale. S is cost penalty for unstructured grid models.

| | S/US | Vertical | Size | Cost v's ORCA25 | | When routine physics model | |
|---|---|---|---|---|---|---|---|
| **Global Scale** | | | (k cells) | No time step, S=1 | S=1 | S=1 | S=3.4 |
| 1/4 | S | 75 | 905 | 1 | 1 | 2011 | |
| 1/12 | S | 75 | 8149 | 9 | 27 | 2017 | |
| 1/36 | S | 100 | 73342 | 108 | 972 | 2023 | |
| 1/72 | S | 100 | 293370 | 432 | 7776 | 2026 | |
| 1/4+1/12 | US | 100 | 2037 | 3 | 9 | 2015 | 2017 |
| 1/12+36 | US | 100 | 10910 | 16 | 145 | 2019 | 2021 |
| 1/12+1/72 | US | 100 | 17409 | 26 | 461 | 2021 | 2023 |
| 1/12+1/72 | BL | 100 | 45558 | 50 | 593 | 2022 | |

S= structured, US = unstructured, BL= blocked refined, e.g. using AGRIF

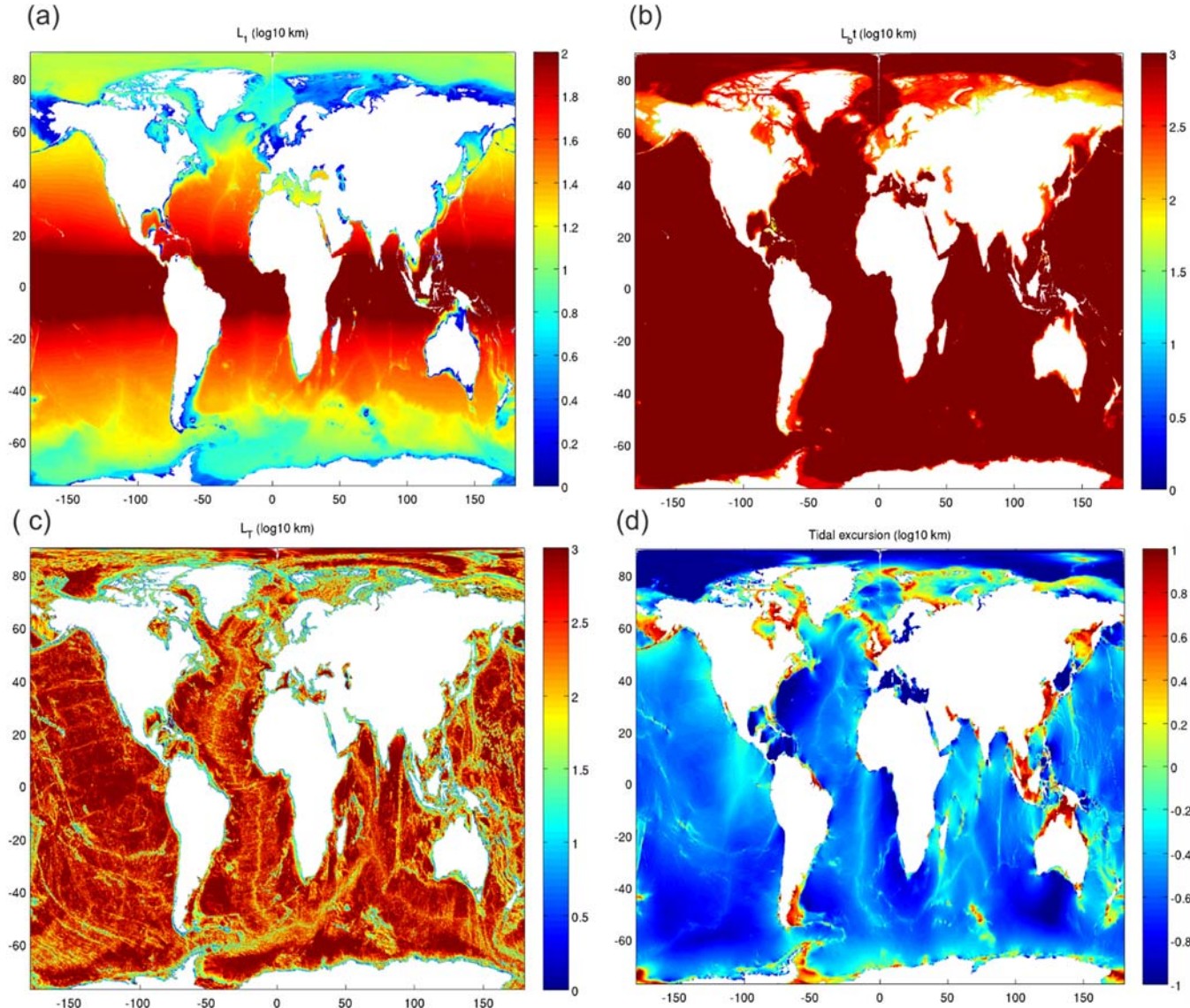

**Figure 1 Global scales: a) 1st Baroclinic Rossby Radius; the maximum value calculated from monthly ORCA12 density profiles (each month being an average from 1981 to 2010) following Nurser and Bacon (2014), using the model run described by Marzochhi et al (2015) ; c) Barotropic Rossby Radius calculated from ORCA12 bathymetry; c) Topographic scale calculated from ORCA12 bathymetry and mesh; d) Tidal excursion, calculated from TPX barotropic tidal currents (Egbert and Erofeeva, 2002).**

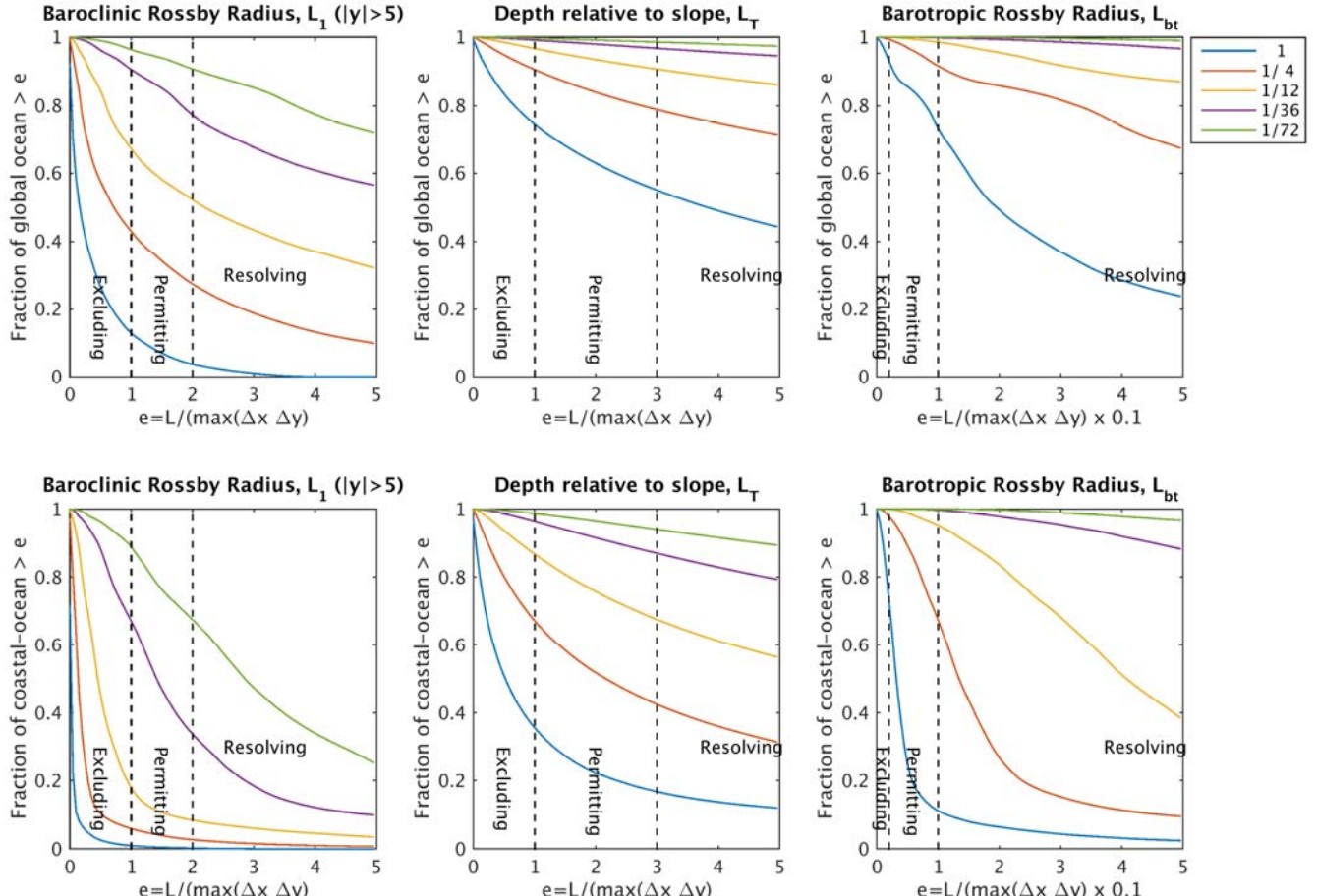

**Figure 2 Cumulative distribution of the fraction of global (top) and coastal (bottom) ocean resolving $L_1$, $L_T$, $L_{bt}$ for different global model resolutions.**

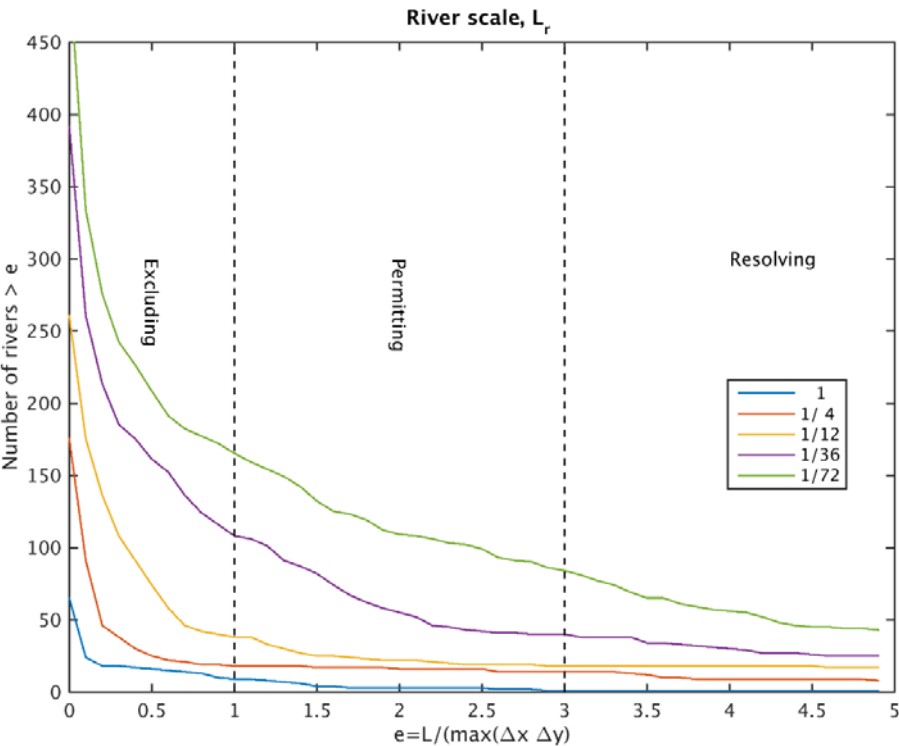

**Figure 3 Cumulative distribution of number of rivers where the scale $L_r$ is resolved at a particular level (e). Based on flow data from the 925 largest ocean-flowing rivers globally (Dai et al., 2009) .**

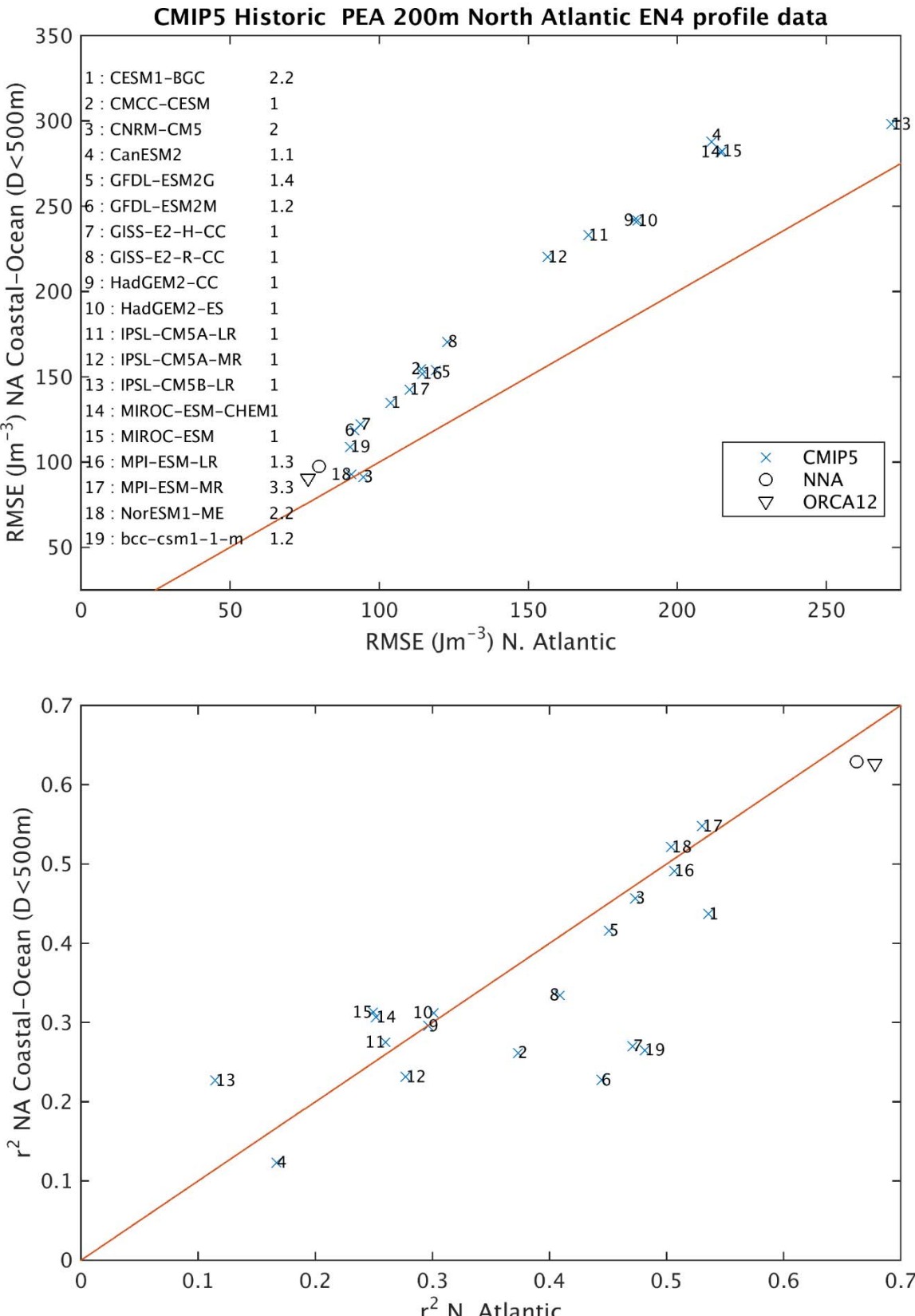

**Figure 4** The RMS error and correlation for PEA of 19 CMIP5 models compared with PEA calculated from EN4 profile data (1970-2014) in the North Atlantic. In both model and observations a mean annual cycle is calculated, and error statistics calculated, with both being interpolated onto the 1/12º Northern North Atlantic (NNA) model grid.  Values for full region are compared with data only at water depths <500m. Values listed by the model names are the inverse mean meridional resolutions of each model. Also shown are results from the global ORCA12 model and from NNA NEMO model (a regional extraction from the ORCA12 grid, with identical vertical coordinates and forced by this at the boundaries) including tides and k-ε (GLS) mixing (both for 1985-2003, DFS forcing).

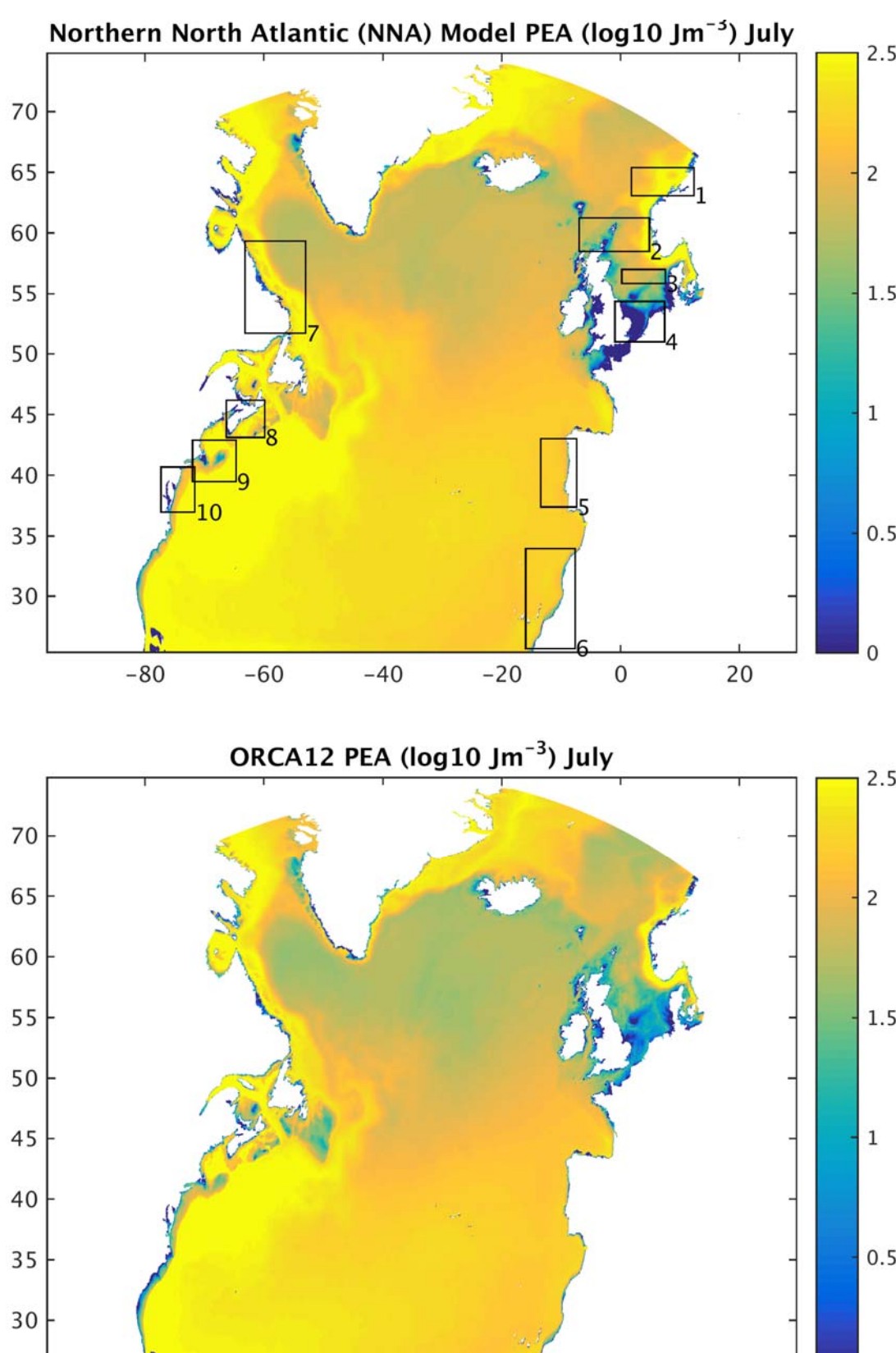

**Figure 5 Potential energy anomaly (PEA; Eqn 2) for July (mean 1985-2003; note log scale) for the Northern North Atlantic (NNA) NEMO configuration (TOP; including tides and k-ε (GLS) mixing) and global ORCA12 model (BOTTOM; with the TKE mixing scheme and no tides). Also shown are regions 1-10 used for a seasonal analysis (Figure 6).**

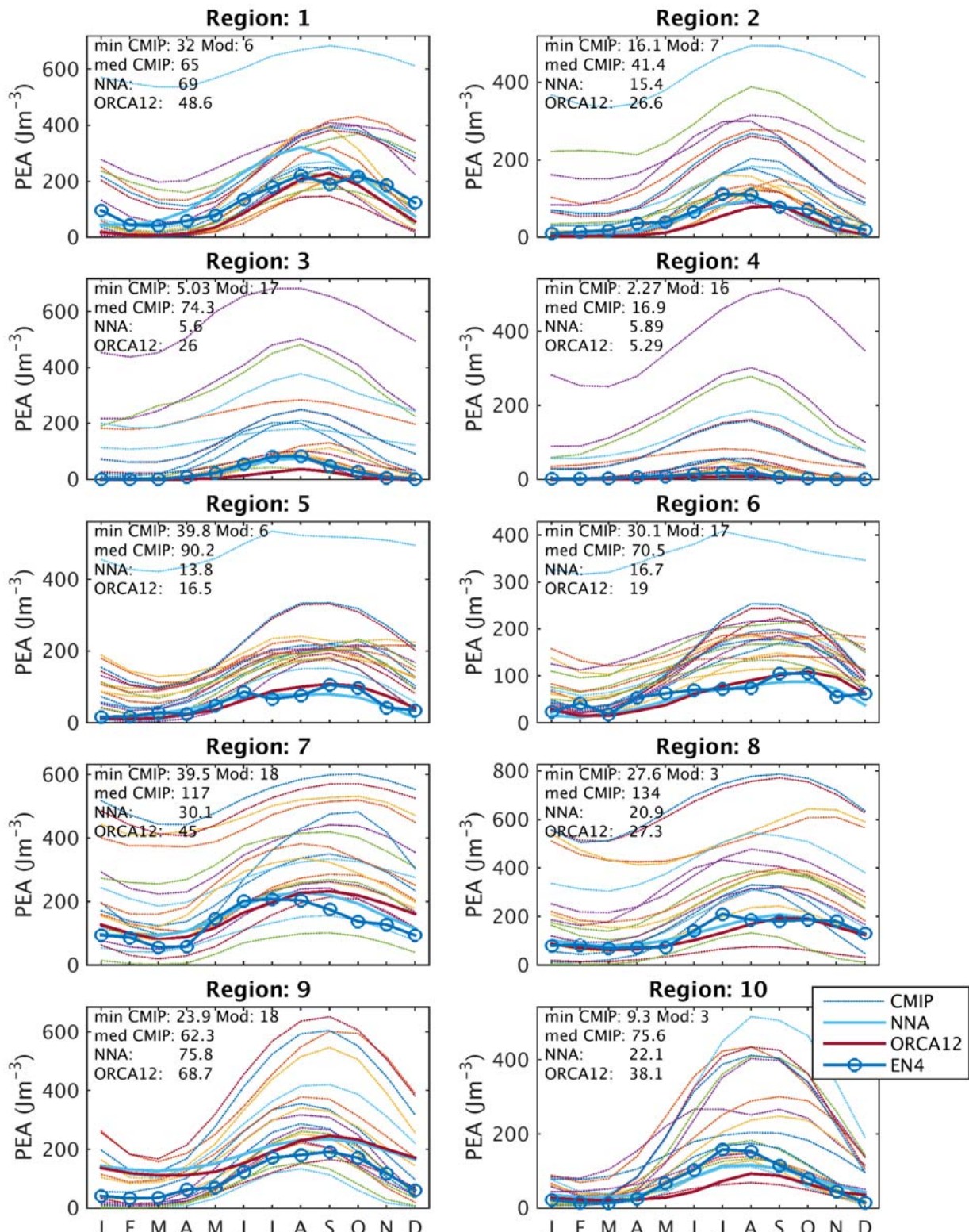

**Figure 6 Mean seasonal cycle of PEA averaged over the 10 regions shown on Figure 5, for water depth <500m. Results for 19 CMIP5 models (light lines) are shown along with the ORCA12, NNA (heavy lines) and EN4 observation (lines and circles). The numbers refer to RMSE compared with EN4 showing the minimum of all these CMIP5 model (and the corresponding model number from the list on Figure 4), the median CMIP5 value and the values for NNA and ORCA12.**

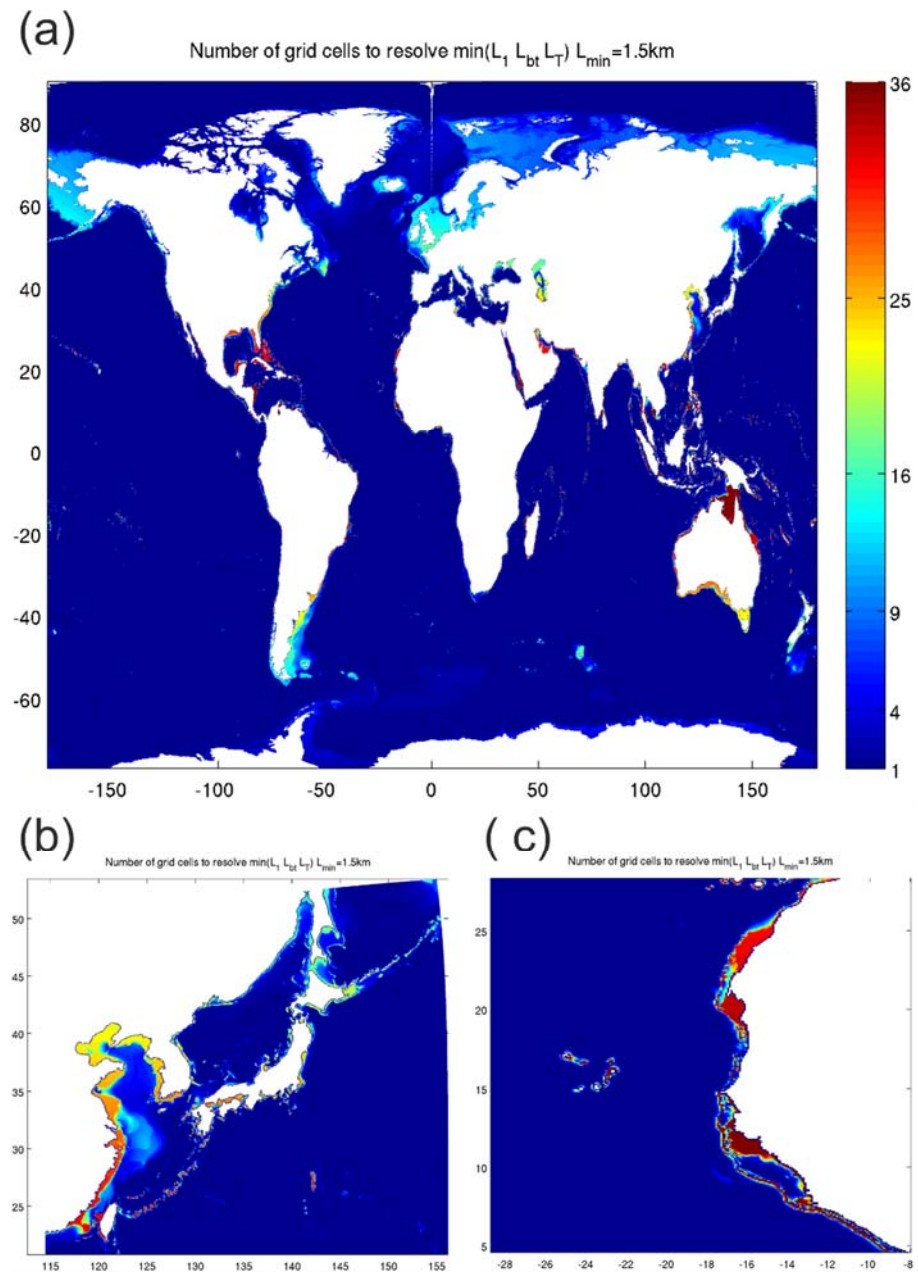

**Figure 7 An example of how a 1/12° global grid (a) might conceptually be refined to resolve the dominant scales. Parameter shown is number of cells needed in each global grid cell to resolve these scales down to a minimum scale of 1.5km, so ranges from 1 (no refinement) to $(72/12)^2=36$. Below are two example in more detail for (b) East Asia and (c) NW Africa.**

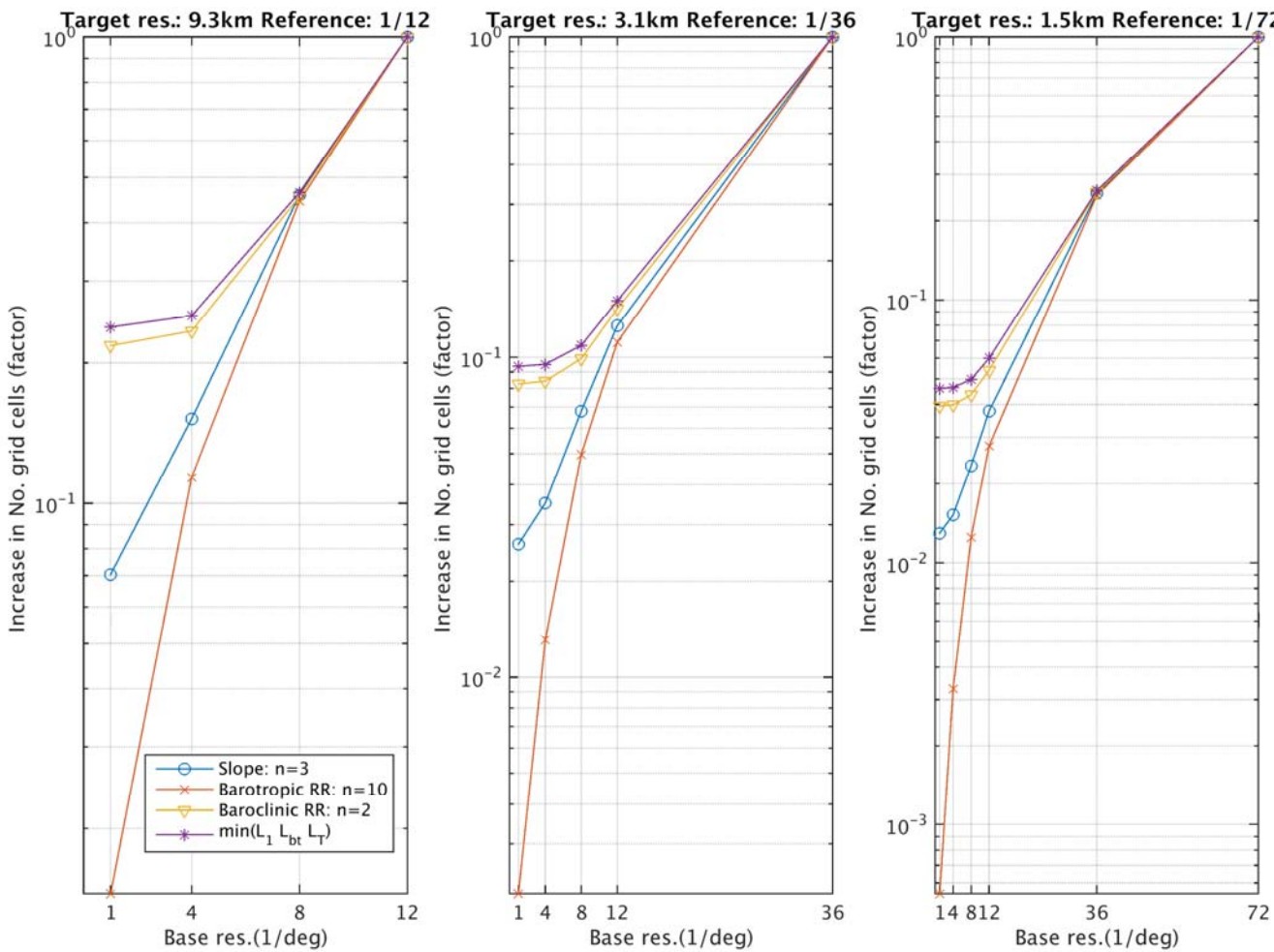

**Figure 8 Number of grid cells to achieve process representation in shelf seas with a multiscale approach, relative to a refining global reference resolution of 1/12 º, 1/36 º and 1/72º and down to a minimum scale set by this global reference.**

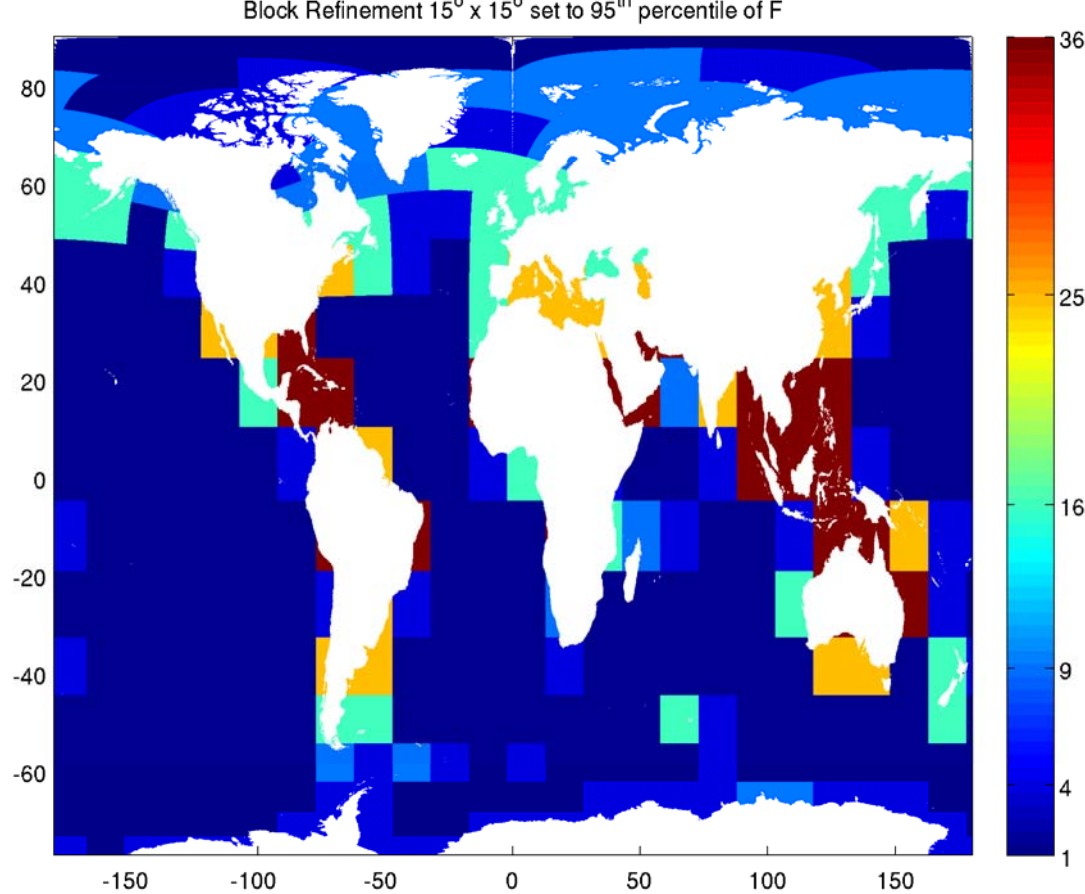

**Figure 9 Refinement of 15°x15° blocks to the 95th percentile of the distribution of $F^2$ in each block. Set to $(int(F))^2$ to approximate refinement by an approach such as AGRIF.**

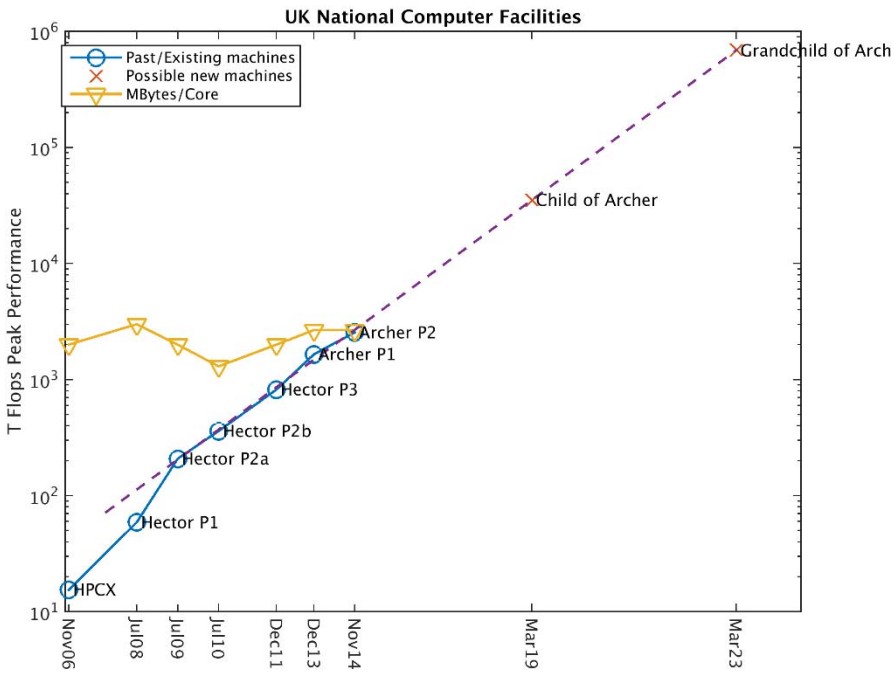

**Figure 10 The UK research computer facility peak performance and memory per core. Also shown are two-projected possible future machines.**