# Peer review of "Prospects for improving the representation of coastal and shelf seas in global ocean models"

_Geoscientific Model Development, 2016_

## Referee Comment (RC1) · S. Danilov (Referee) · 31 Jul 2016

The manuscript discusses aspects of coastal versus global ocean modeling. Its basic conclusions rest on estimates of ocean scales. These estimates are of interest, but they are insufficiently deep. The technical side of the problem of matching coastal and global scales in a single setup is addressed from the side of mesh resolutions. Many physical aspects are mentioned, but without practical recommendations.

While it is true that a precise boundary between coastal and global scales does not exist and a lot will be possible as resolution is further refined, there are even smaller scales or physical processes resolving which in a global setup will make the global approach strongly suboptimal (for example, regimes with wetting and drying). So I see the technologies based on two-way nesting or unstructured meshes much more

promising and requiring much less computational resources in many coastal tasks. I would recommend to define more precisely the context of global coastal modeling. I would also recommend to put more focus on specific issues such as the reduction of spurious mixing on terrain following meshes as suggested by Lemarie et al. (2012, Ocean Modelling), the Arbitrary Lagrangian–Eulerian vertical coordinate (MPAS and MOM6), scale-aware mixing and eddy parameterizations, measures influencing numerical stability if the intention is to really discuss the prospects.

The analysis of future perspectives based on the evolution laws for the available computational power is a bit superficial. The point is not only the computational power on its own, but also time step limitations. The manuscript considers additional resources needed because the time step will decrease on finer meshes. However, if the scalability stays as its present level, the codes will become slower in terms of simulated years per day of computations. If a 1/72 degree model runs slower by a factor of 6 (because of the time step reduction) than a 1/12 one, the question is how many days it will take to run such a model over a cycle of CORE-II (or any other) forcing which is a bare minimum for global simulations. Any prognosis remains vulnerable to the issues of scalability and parallel efficiency which are difficult to estimate. The key question is which technologies can improve scalability and how, and without answering it the rest is just an exercise. Running 1/10 or 1/12 degree models is affordable today to many groups, but such models remain too slow for exploring climate over large time intervals.

The view on unstructured meshes proposed in the manuscript seems to be outdated and is based on just arbitrary comparison for a particular model (FVCOM). Here, the point is better parallel scalability of current unstructured-mesh codes on large meshes which warrants comparable throughput (but with somewhat larger demand to computer resources per degree of freedom). Furthermore, unstructured-mesh codes deal only with wet nodes, which creates noticeable economy in coastal areas. For example, even with FESOM1.4, which is relatively slow because of its 1D arrays, one reaches throughput of about 8 simulated years per day on global locally eddy-resolving meshes that are

as large or larger than the 1/4 degree mesh (see Sein et al., 2016), which is not worse than the performance of regular-mesh models on 1/4 degree meshes. New developments such as MPAS, FESOM2 and ICON are a factor 2 to 3 faster than FESOM1.4 and can compete with structured-mesh models in massively parallel applications.

If the coastal area occupies less than 10%, running an unstructured-mesh model or model with two-way nests on a mesh that is three times coarser in the global ocean than in the coastal part is 5 times less expensive (in terms of computational resources) than running this model on a global fine mesh. Such arithmetic is trivial and hardly tells in favor of global fine meshes; devoting a significant place in the manuscript to all the would be issues is not very appropriate (for it depends on the slowness factor of 5 that the authors took for unstructured meshes).

Based on these remarks I would recommend a major revision. I would be happy to see more substance. The manuscript presents a view, but does not formulate solutions which respect of numerics of physics one expects to see in a GMD paper.

Below are more specific comments.

Background and motivation:

The contents of 1.2 and 1.3 are a bit shallow to warrant their publication. 1.3 fails to convey a message that downscaling cannot be done with nesting, for one just needs to take a (two-way) nest of appropriate size. Examples of 1.2 are well known, but the point is not only the resolution in the ocean, it is also bottom representation, spurious mixing etc. So it is not only upscaling. The manuscript does not really discuss measures needed for seamless representation of particular processes. This all leaves me wondering why 1.2 and 1.3 are needed at all? Most readers of this journal already know it, and I would omit them.

Page 3, lines 3-5: The sills between the Nordic seas and the North Atlantic are not as deep as 1500-3000 m. So the NADW is very indirectly connected to what is said.

Page 4, bottom, the first and third issues are rather close. Page 5: top, rivers are also accounted for in global models Fast ice is accounted (through parameterization) in some large-scale models; it is not a strong point of most coastal models.

The beginning of 2.1: To compute the scales one does not need a NEMO grid and can use the original data instead. So you need it to put the scales into the context of particular model. There is no issue of North Pole singularity for scalar data.

Page 6: If L1=c/f, where c is the speed of the first mode, then for the Eady instability wave the wavelength of most unstable wave is approx. 3.9 \pi L1, giving different size of eddies.

Page 8, top: Mentioning barotropic tidal models is hardly relevant in the context of 3D modeling. Instead of reviewing who did what it would be much more appropriate to formulate what is the essence of difficulties and only then discuss the current status.

Discussion of spurious mixing in the context of tides sounds strange to me — according to Ilicak et al (2013) it is the grid scale motions that have the largest impact. The point of spurious mixing deserves much more attention, for it is also related to terrain following coordinates which intersect isopycnals at some angle and introduce spurious mixing precisely where they are most needed (the continental break boundary). The manuscript mentions it, but not the measures needed to overcome the difficulties.

Page 9: Transition to terrain-following coordinates in shallow water: This option is available in some other models (SELFE, FESOM) from much earlier date.

The message of Figure 5 is not very clear. It does not show why or which technology should be selected.

Page 10: "Mixing of temperature and salinity usually takes place along isoneutral, rather than geopotential, surfaces and this requires careful implementation in the case of sloping vertical coordinates." What about transport algorithms with upwinding or limiters ? Most coastal applications will need upwinding and limiters.

Line 20: The simplest is of course scaling with the horizontal mesh cell size for harmonic and cube of that for biharmonic operators as is routinely done in most codes. Discussion mixes horizontal (Smagorinsky) and vertical aspects. Scale-selective approaches deserve more attention. In most cases the horizontal subgrid operators just aim to remove the grid-scale variance, which is far from physically motivated solutions.

Line 25: "Quadrilateral meshes approximate coastlines by imposing zero-normal flow condition on specific edges of the mesh and masking the landward solution." — I think all meshes do the same in this respect.

The representation of coastlines should not be a big issue on terrain following quadrilateral meshes if the depth tends to zero. On the other hand, on triangular meshes smooth coastlines do not really help unless terrain-following meshes are used, for boundary of any layer should be smooth. If wetting and drying is present, then again it is the bottom representation. So there is general problem of bottom representation, and the example of Kelvin wave is in essence related to it. The representation of coastlines is a part of this general problem, and cannot be considered separately.

Section 3.2. ORCA meshes assume certain scaling (largely with latitude), so together with calculations for them it would be of interest to present a mesh-independent calculations: you have your pattern of the smaller of the baroclinic, barotropic, and topographic scales and can compute distribution of mesh nodes over scales for just resolving and for resolving with two points per scale. This is more informative for unstructured meshes. A delicate question is the behavior of time step with resolution, which needs some extra discussion. ORCA meshes partly account for the reduction of the phase speed of internal waves with latitude, but other processes may be (locally) limiting at fine scales. The discussion of the number of mesh points looks unsatisfactory (it is elementary) to me if not augmented by time step analysis, at least on a qualitative level. Scale $L\_min$ may imply different time step selection at different locations, so the question is what is the optimal strategy.

Section 3.3.

Page 11, line 40: In reality hybrid approach is used by all these models as concerns the computation of pressure gradients. It is also true for many other models.

Page 12: Line 1: I think the authors derived a wrong message. There is no point with formal accuracy on unstructured meshes, in fact smooth triangular meshes are more isotropic than quadrilateral and will be more accurate. Computational modes indeed require attention, but there are solutions how to handle them. The technology is mature enough, it requires more caution.

Line 3: Wave propagation on quadrilateral meshes has no advantages. Quadrilateral meshes are simply cheaper for there are less edges, which is crucial for finite-volume codes.

Line 10: "The former ....but have not yet reached ...." Please be careful, for the statement is wrong. FESOM, for example, was a part of CORE-II intercomparison (see the virtual special issue of Ocean Modelling), it is a component of AWI climate model (see Sidorenko 2015, Clim. Dyn), and in this way participates in CMIP6.

I think that the unstructured-mesh part of this section is weak and does not convey a correct message to community.

Section 4.

The discussion proposed in this section misses some important points. If the clock speed peak is already reached, the throughput of codes on fine meshes will decrease on mesh refinement. This aspect is not less practical than the availability of computational power and has to be mentioned. The authors state "This requires at least three-way nested parallelism ..." Are there solutions allowing to efficiently work with smaller mesh pieces per core? A perspective in this direction should be addressed. The other aspect is structured vs unstructured codes. In finite-volume unstructured-mesh codes the neighborhood information is two-dimensional, so it is related to the vertical column

of computational points and accessing it is not expensive. Computations of high-order advection or gradients are more expensive than on sttuctured meshes, but if memory bandwidth is a limiting factor, the need in more computations will be less apparent. Furthermore, mesh partitioning is easier (it is derived from the connectivity pattern and involves only wet nodes). I am not sure whether discussing all this is possible in this manuscript, but proposing some discussion would provide a much more valuable message to the community.

The statement on page 15 "Hence, unless very efficient methods of multiscale modelling are developed ....." can be critisized, and the extent of this depends on what we define as efficiency and what oceanographic task we are considering. I would not do here, but only note that multiscale modeling methods can be (and are) much more efficient than assumed in the manuscript.

I think Fig. 10 is not really necessary, for much more work is needed to explore functioning of such meshes for eddy-rich dynamics. There is no substance at present. It is in contrast to numerous other efforts on triangular meshes which are barely addressed.

S. Danilov

---

## Referee Comment (RC2) · R. Proctor (Referee) · 6 Aug 2016

The paper examines the state of the art in the representation of shelf seas in global models and explores options for their better representation in future in terms of resolution, processes and numerical schematisation mapped against potential increases in compute resource.

The paper provides a background context by considering the interaction between shelf seas and the ocean in terms of impacts, i.e. upscaling and downscaling, and in terms of process representation. The bulk of the paper considers physical process resolution in terms of a set of characteristic length scales and assesses the relative 'cost' of refining processes to capture coastal and shelf sea scales in global models. The final section considers the cost benefit of three different numerical schematisations for capturing

coastal and shelf scales in global models – quasi-uniform structured meshes, unstructured meshes, multi-blocking refinement – against projected computer resource.

The 'standard' model against which the various options are considered is the 1/12 degree NEMO global model. The scale considered acceptable for coastal and shelf processes is a few metres in the vertical and 1.5km in the horizontal, leading to the requirement of at least 1/72 degree resolution in coastal and shelf sea regions and some form of hybrid coordinate in the vertical. It is suggested that such a requirement could be met in 6-10 years depending on the numerical schematisation employed.

Three points for me make revisions to the paper necessary:

1) Most of the introductory theory is well-known to modellers in both ocean and shelf communities, and could be truncated, keeping the informative figure 1 and reducing figures 2 and 3 since there is no discussion of 1/24 and 1/48 degree resolution, i.e. they are unnecessarily complicated. The same can be said for figure 7 as 1/36 degree resolution is only briefly discussed. I also feel that figures 4,5 and 6 contribute little enhancement over the given text and can easily be omitted.

2) It is understood that most of the authors are practised users of quasi-uniform structured grids, and, understandably, take one of their models as the benchmark. However the treatment of other numerical schematisations is given only cursory consideration and selects, what seems to me, to be a totally arbitrary scaling factor (5) for the performance of unstructured models (in this case FVCOM is chosen) against NEMO in a limited shelf region (without any detail of these simulations), which is then applied on a global scale. A more considered view of alternative numerical representations is warranted, and the authors are aware of some of these (e.g. FESOM, MPAS) and there are others (e.g. SELFE, SCHISM), and what about adaptive grid schemas? Also, there is no discussion of the efficiency of NEMO against other comparable models which are global (e.g. MOM, HYCOM, ROMS).

3) It is probable that, if aiming for a coastal-ocean resolving global model (i.e. 1/72

degree), then some of the processes discussed e.g. tides, sea ice, etc. will need to be included to properly represent the physical processes in these regions at this scale, so some estimates of their inclusion ought to form part of the discussion.

A few minor points, but ones that need attention:

Page 2, line 23. Liu et al reference missing

Page 5, line 14. Robinson & Brink reference missing

Page 5, line 17 "...tides are ubiquitous in the coastal-ocean". Well, no they are not an important process to resolve everywhere, some coastal ocean regions have extremely small tides.

Page 6, line 34. Please better explain the factor E

Page 7, line 35. Madec reference missing

Page 8, line 29. St. Laurent reference missing

Page 9, line 20+. Worth adding further detail regarding the sophistication of the hpg calculation and explanation of its impact on the energy cascade

Page 10, line 12. "... but is not generally used ..." is it 'used' or 'required'?

Page 12, line 5. States Figure 9, should be Figure 10

Page 12, line 8. Need a reference for the statement "serious numerical issues"

Page 12, lines 13-16. Sentence badly constructed

Page 13, line 30. What are the implications of this energy figure?

Page 13, line 31. The projections of compute power in Figure 9 appear to be based on extrapolations of existing architecture, how realistic is this assumption?

Page 14, line 38. Spelling ... ocean.

Page 17, line 9. Bryan reference incomplete.

---

## Author Comment (AC1) · 28 Sep 2016

We thank Dr Danilov for his careful consideration of this paper and his helpful suggestions on how to improve it. He has some concerns, which we will aim to address in a revised version. The general criticism raised is that the work is insufficiently deep and lacks practical recommendations with respect to numeric approaches. The aim of this paper is to clearly articulate the problem, assess possible solutions and how practical they are on particular time scales. In the revision, we aim to improve the depth of this analysis by being more quantitative in our assessment, particularly by drawing more on the observational base (e.g. fig 4), how the current generation of global models (e.g. CMIP5 and where we have data new CMIP6 models) performs against this in the coastal regions and, whether the developments suggested (in fig 5) improve on this, using a direct comparison with observations. It is not the objective of this paper to offer

new algorithms to provide solutions to this problem. Describing specific, novel, numerical solutions to the issues presented is substantially beyond the scope of this work and could not be addressed in a single paper. We can, however, be more detailed in our description and assessment of solution options in the literature, and this will be a key aspect of the revision. He suggests several aspects that deserve greater attention, namely reducing spurious mixing, factors effecting stability and scale-aware mixing parameterisation. We agree these are all important and will ensure they are appropriated covered in the revision.

Dr Danilov very helpfully points out that we neglect the important, related, issues of time stepping, scalability and throughput, and this will be addressed in the revision. Essentially we focus on resource (cpu.hrs), rather than time to complete a simulation (throughput). The former is what is metered by our computer centres, but the latter limits how much science we can do with the resource. As grids are refined, throughput will reduce irrespective of available resource unless scalability can be approved (fewer grid cells efficiently allocated to each MPI process) or time stepping made more efficient. We will consider if these can be included in our cost model, and will add to the discussion.

Dr Danilov is correct that our considerations of unstructured mesh models is somewhat out of date, and this will be addressed in the revision. Particularly we will look in the literature for better estimates for the addition costs of unstructured v's structured models to improve on the factor of 5 we use in our cost model. We will also consider in more detail what the factors affecting this ratio are (complexity of code, indirect memory addressing etc), and how they can be alleviated. It is not our intention to disparage unstructured mesh approaches here, but rather make a balanced, realistic assessment.

The more specific comments, which we generally agree with (with some minor exceptions), will all be addressed in the revision. Particularly sections 1.2 and 1.3 will be combined and reduced, with aim of making our motivation clearer. We agree that

figure 10 and the discussion around it can be deleted.
* * *

---

## Author Comment (AC2) · 28 Sep 2016

We thank Dr Proctor for his careful consideration of this paper and his helpful suggestions on how to improve it. We agree that sections 1.2 and 1.3 can be shortened and figure 2 simplified. However, we feel figures 4 and 5 provide important information but are not so well exploited. In the revision we will aim to use the data from fig. 4 to validate the runs in figure 5 to demonstrate (or otherwise) that these coastal-ocean features (e.g. tides) improve the simulations. We may also be able to use this data to assess other current generation global ocean models (e.g. from CMIP5) in the coastal ocean. Figure 6 is important in illustrating the method of calculating the size of the multiscale models – without it figure 7 would be difficult to interpret. But we could combine 6 and 7 into a single figure.

[Figure]

We agree unstructured mesh modelling is discussed in insufficient detail in the paper. In the revision we will consider more recent approaches in greater detail, but will stop short of a comprehensive review as this is outside the scope of the present paper. We will focus on issues of time-stepping, scalability and efficiency, looking for evidence in the literature as to what these might be. The scaling factor currently used in the paper reflects our present experience with regional nemo and fvcom models. Dr Proctor is most probably correct in asserting that this may well not be appropriate for a global analysis, so we will aim to provide a more robust estimate or omit it if this is not possible. In the wider considerations of scalability and efficiency, we will aim to go beyond NEMO to see how it compares with other structured grid models as suggested. But this comparison may be limited by the available evidence.

Finally, regarding the estimate of including additional coastal ocean processes (such as tides and sea-ice) on model costs, we will draw on our experience with the Northern North Atlantic model (fig 5) to inform a discussion on this.

The other more detailed and minor comments will be dealt with on a case by case basis.

---

## Author Response (AR1)

Dear Bob, we have revised the manuscript to comprehensively address the reviewers' comments, within limits of what is practical and to maintain the original scope and aim of the paper. We hope it is now acceptable for publication,

Best wishes,

Jason

**Prospects for improving the representation of coastal and shelf seas in global ocean models**

**Response to referee: RC1**

*These estimates are of interest, but they are insufficiently deep.… I would be happy to see more substance. The manuscript presents a view, but does not formulate solutions which respect of numerics of physics….*

It was never our intention to provide a comprehensive solution to this question with specific numerical options. This would essentially require the description of a new model. Instead we aim in this revision to better articulate the challenge, and so offer more 'substance'. To this end, we now include a new comparison of CMIP5 models with an observed profile data set and the NNA and ORCA12 high resolution NEMO models, which we believe is novel, robust and informative. This now clearly demonstrates some of the benefits of improved resolution and process representation. P6 L25 and Section 3.2. Figs 4, 5, 6.

*The technical side of the problem of matching coastal and global scales in a single setup is addressed from the side of mesh resolutions. Many physical aspects are mentioned, but without practical recommendations.*

We have developed section 3.2 with a list of specific recommendations and issues, drawing on the literature.

*While it is true that a precise boundary between coastal and global scales does not exist and a lot will be possible as resolution is further refined, there are even smaller scales or physical processes resolving which in a global setup will make the global approach strongly suboptimal (for example, regimes with wetting and drying). So I see the technologies based on two-way nesting or unstructured meshes much more promising and requiring much less computational resources in many coastal tasks. I would recommend to define more precisely the context of global coastal modeling.*

We agree regional models with both structured and unstructured mesh approaches remain the optimal solution for many coastal-ocean questions. To this end we better articulate the drivers for using global models in the coastal-ocean in a revised section 1.1, which ends with a list of the specific types/uses of global model this work is aimed at (P3 L28). One practical point that should not be neglected, but was not made in the original submission is the cost (e.g. in scientists time) of configuring and testing a new regional model for each area of interest. If readily available global information had improved coastal-ocean representation then this could substantially aid a diverse range of activities, albeit often sub-optimally to setting up a new region configuration for each case, but particularly useful for the research community who are not experts in coastal ocean modelling (P3 L20-27).

*I would also recommend to put more focus on specific issues such as the reduction of spurious mixing on terrain following meshes as suggested by Lemarie et al. (2012, Ocean*

*Modelling), the Arbitrary Lagrangian–Eulerian vertical coordinate (MPAS and MOM6), scale-aware mixing and eddy parameterizations, measures influencing numerical stability if the intention is to really discuss the prospects.*

We agree these are very important issues and have developed the discussion of these, section 3 (P8 L25 onwards). But we are not in a positon to offer new solutions here or further test existing ones.

*The analysis of future perspectives based on the evolution laws for the available computational power is a bit superficial. The point is not only the computational power on its own, but also time step limitations.*

*And..*
*However, if the scalability stays as its present level, the codes will become slower in terms of simulated years per day of computations. The key question is which technologies can improve scalability and how, and without answering it the rest is just an exercise.*

Given the many uncertainties of the evolution of computing we do not feel it appropriate to consider this in more detail than simply extrapolating the current tread in peak performance (e.g. to look at projections of future chip technology).
The time step question is a crucial point and we have added a section on this (P13 L34-P14 L2). We have also added a section on ocean model performance and scalability (section 4.4 P15), and identified one practical way to re-code ocean models to make optimization on new computer architectures much more straightforward. This has the potential to substantially improve ocean model scalability (P16 L24 onwards).

We now make the important distinction between resource and turn-around time and provide some specific re-world examples of realised SYPD (P15 L35; Table 3). Some ocean modelling activities will be resource limited, others SYPD limited. Including scalability into the simple cost model we use is not straightforward and would complicate this beyond what is warranted, given the other uncertainties. So we leave it as a simple estimate of resource needed, but make clear the SYPD may decrease even if more resource is available.

*The view on unstructured meshes proposed in the manuscript seems to be outdated and is based on just arbitrary comparison for a particular model (FVCOM). Here, the point is better parallel scalability of current unstructured-mesh codes on large meshes which warrants comparable throughput (but with somewhat larger demand to computer resources per degree of freedom).*

Agreed and we have revised section 3.4 accordingly, reducing the discussion on unstructured meshes to fit with the original assertion that we would not try and review these here. We have added a discussion on the relative performance of structured and unstructured meshes (P15 L35), related to the real-world examples in Table 3. On this basis we now give a range of figures allowing for the resource-use of unstructured mesh models to vary from about 3 to 1 (parity with structured mesh models). In this way we feel we now present a more balanced view.

*Furthermore, unstructured-mesh codes deal only with wet nodes, which creates noticeable economy in coastal areas.*

This maybe important for coastal models with complicated topography (e.g. estuaries) but is a marginal issue for global models run on many 1000's of processors and where land-only processors are excluded (P15 L41).

*If the coastal area occupies less than 10%, running an unstructured-mesh model or model with two-way nests on a mesh that is three times coarser in the global ocean than in the coastal part is 5 times less expensive (in terms of computational resources) than running this model on a global fine mesh. Such arithmetic is trivial and hardly tells in favor of global fine meshes; devoting a significant place in the manuscript to all the would be issues is not very appropriate (for it depends on the slowness factor of 5 that the authors took for unstructured meshes).*

The arithmetic may be simple but providing an objective basis for the 'ocean area occupies less than 10%' and the global ocean is 'three times coarser' is far from straightforward, and this is central to the question we are trying to address. We could add other layers of complexity to the calculation – e.g. by assuming a maximum gradient of grid-refinement, but we do not feel these would helpfully inform the overall message. We do agree the 'factor of 5' needed to addressed and have modified as described above.

Other points
*The contents of 1.2 and 1.3 are a bit shallow to warrant their publication. 1.3 fails to convey a mesage that downscaling cannot be done with nesting, for one just needs to take a (two-way) nest of appropriate size. Page 3, lines 3-5: The sills between the Nordic seas and the North Atlantic are not as deep as 1500-3000 m. So the NADW is very indirectly connected to what is said.*

These sections have been substantially edited and reduced in size to now just convey the overall motivation with some brief examples.

*Page 4, bottom, the first and third issues are rather close. Page 5: top, rivers are also accounted for in global models Fast ice is accounted (through parameterization) in some large-scale models; it is not a strong point of most coastal models.*

True, but this section refers to properties of these regions, irrespective of how they are treated in the current generation of models (P4 L11).

*The beginning of 2.1: To compute the scales one does not need a NEMO grid and can use the original data instead. So you need it to put the scales into the context of particular model. There is no issue of North Pole singularity for scalar data.*

True – data here originated on the NEMO grid (apart from TPX) so it is appropriate to use this grid. An area-weighted mean is used, so the results are not dependent on this choice of grid (P5 L30). The mention of a NP singularity has been removed.

*Page 6: If L1=c/f, where c is the speed of the first mode, then for the Eady instability wave the wavelength of most unstable wave is approx. 3.9 npi L1, giving different size of eddies.*

We have changed to use this measure of eddy scale and re-drawn fig 2 accordingly.

*Page 8, top: Mentioning barotropic tidal models is hardly relevant in the context of 3D modeling. Instead of reviewing who did what it would be much more appropriate to formulate what is the essence of difficulties and only then discuss the current status.*

Agreed and the text on barotropic tidal models has been removed.

*Discussion of spurious mixing in the context of tides sounds strange to me—according to Ilicak et al (2013) it is the grid scale motions that have the largest impact. The point of spurious mixing deserves much more attention, for it is also related to terrain following coordinates which intersect isopycnals at some angle and introduce spurious mixing*

*precisely where they are most needed (the continental break boundary). The manuscript mentions it, but not the measures needed to overcome the difficulties.*

Spurious mixing in the context of tides is likely to arise when large amplitude internal tides are generated. Without special treatment coordinate surfaces will only move with the barotropic tide, so the baroclinic tide will lead to high frequency isopycnal displacement across the coordinate surface. This is likely to lead to spurious vertical mixing. It is the motivation for the development of the z-tilda coordinate in NEMO. The discussion on this has been expanded (P7 L37), although we are not in a position to offer novel solutions here.

*Page 9: Transition to terrain-following coordinates in shallow water: This option is available in some other models (SELFE, FESOM) from much earlier date.*

This has been noted (P9 L4).

*The message of Figure 5 is not very clear. It does not show why or which technology should be selected.*

This figure has been replaced with a quantitate comparison with CMIP5 models and observations, in both the overall (Fig 4) and seasonal cycle (Fig 6) cases, alone with a qualitative comparison between NNA and ORCA12 (Fig 5). This provides a much clearer message on the benefits of improved resolution and process representation.

*Page 10: "Mixing of temperature and salinity usually takes place along isoneutral, rather than geopotential, surfaces and this requires careful implementation in the case of sloping vertical coordinates." What about transport algorithms with upwinding or limiters? Most coastal applications will need upwinding and limiters.*

It not clear what is being suggested here. The NEMO model uses the same transport algorithms in coastal and open-ocean applications, which is appropriate.

*Line 20: The simplest is of course scaling with the horizontal mesh cell size for harmonic and cube of that for biharmonic operators as is routinely done in most codes.*
Agreed

*Discussion mixes horizontal (Smagorinsky) and vertical aspects.*
This has been clarified with section headings
.
*Scale-selective approaches deserve more attention.*

They do deserve more attention, but we are not in a positon to provide any further developments on this here and are not aware of any other published work in this area.

*In most cases the horizontal subgrid operators just aim to remove the grid-scale variance, which is far from physically motivated solutions.*

Agreed and noted (P9 L35)

*Line 25: "Quadrilateral meshes approximate coastlines by imposing zero-normal flow condition on specific edges of the mesh and masking the landward solution." — I think all meshes do the same in this respect.*

Agreed and corrected P10 L13.

*The representation of coastlines should not be a big issue on terrain following quadrilateral meshes if the depth tends to zero. On the other hand, on triangular meshes smooth coastlines do not really help unless terrain-following meshes are used, for boundary of any layer should be smooth. If wetting and drying is present, then again it is the bottom representation. So there is general problem of bottom representation, and the example of Kelvin wave is in essence related to it. The representation of coastlines is a part of this general problem, and cannot be considered separately.*

This may be true, but the depth does not generally tend to zero in terrain following coordinate models due to the combatively coarse resolution considered (compared with coastal models of 100m's meter resolution; P10 L25), particularly when wetting-drying has not been implemented and a minimum depth is required. This implies that Kelvin-wave retardation remains a potential issue for the barotropic solution of structured mesh models where the coast is not aligned with the coordinate system. The relation between this and the underlying topography is now discussed (P10 L18).

*Section 3.2. ORCA meshes assume certain scaling (largely with latitude), so together with calculations for them it would be of interest to present a mesh-independent calculations: you have your pattern of the smaller of the baroclinic, barotropic, and topographic scales and can compute distribution of mesh nodes over scales for just resolving and for resolving with two points per scale. This is more informative for unstructured meshes.*

Agreed it would be possible and maybe interesting to reproduce this refinement process with other underlying mesh structures, but we're not sure this significant added complication (e.g. extra figures and description) would add much to the paper.

*A delicate question is the behavior of time step with resolution, which needs some extra discussion. ORCA meshes partly account for the reduction of the phase speed of internal waves with latitude, but other processes may be (locally) limiting at fine scales.*
*And…*
*The discussion of the number of mesh points looks unsatisfactory (it is elementary) to me if not augmented by time step analysis, at least on a qualitative level.*
*And…*
*Scale L_min may imply different time step selection at different locations, so the question is what is the optimal strategy.*

Agree and we have added a section on timestepping (P13 L34). In all the resource estimates we have assumed a single global time step and this scales with the inverse of the minimum length scale, now explicitly stated: P17 L9. An exception is the block refined case, where we assume each refined block can have a different time step and appropriate load balancing is in place (P17 L18). We mention the possibility of locally varying timesteps in discussion (P14 L1) but do not include this possibility in our resource estimates for unstructured mesh models.

*Page 11, line 40: In reality hybrid approach is used by all these models as concerns the computation of pressure gradients. It is also true for many other models.*

Noted – this has been deleted.

*Page 12: Line 1: I think the authors derived a wrong message. There is no point with formal accuracy on unstructured meshes, in fact smooth triangular meshes are more isotropic than quadrilateral and will be more accurate. Computational modes indeed require attention, but there are solutions how to handle them. The technology is mature enough, it requires more caution*

Noted and no longer included

*Line 3: Wave propagation on quadrilateral meshes has no advantages. Quadrilateral meshes are simply cheaper for there are less edges, which is crucial for finite-volume codes.*

Noted and no longer included.

*Line 10: "The former ....but have not yet reached ...." Please be careful, for the statement is wrong. FESOM, for example, was a part of CORE-II intercomparison (see the virtual special issue of Ocean Modelling), it is a component of AWI climate model (see Sidorenko 2015, Clim. Dyn), and in this way participates in CMIP6. I think that the unstructured-mesh part of this section is weak and does not convey a correct message to community.*

Section 3.4 has been re-written to address these concerns.

*The discussion proposed in this section misses some important points. If the clock speed peak is already reached, the throughput of codes on fine meshes will decrease on mesh refinement. This aspect is not less practical than the availability of computational power and has to be mentioned. The authors state "This requires at least threeway nested parallelism ..." Are there solutions allowing to efficiently work with smaller mesh pieces per core? A perspective in this direction should be addressed. The other aspect is structured vs unstructured codes. In finite-volume unstructured-mesh codes the neighborhood information is two-dimensional, so it is related to the vertical column of computational points and accessing it is not expensive. Computations of high-order advection or gradients are more expensive than on sttuctured meshes, but if memory bandwidth is a limiting factor, the need in more computations will be less apparent.*

*Furthermore, mesh partitioning is easier (it is derived from the connectivity pattern and involves only wet nodes). I am not sure whether discussing all this is possible in this manuscript, but proposing some discussion would provide a much more valuable message to the community.*

We now provide a section discussing the efficiency and scalability of ocean models (section 4.4) and provide a practical suggestion on how they can be coded to accommodate this three-way nested parallelism (P16 L24).

*The statement on page 15 "Hence, unless very efficient methods of multiscale modelling are developed ....." can be critisized, and the extent of this depends on what we define as efficiency and what oceanographic task we are considering. I would not do here, but only note that multiscale modeling methods can be (and are) much more efficient than assumed in the manuscript.*

This has been corrected both in the text and resource estimations.

*I think Fig. 10 is not really necessary, for much more work is needed to explore functioning of such meshes for eddy-rich dynamics. There is no substance at present. It is in contrast to numerous other efforts on triangular meshes which are barely addressed.*

Agreed and this has been removed.

**Response to RC2**

*Most of the introductory theory is well-known to modellers in both ocean and shelf communities, and could be truncated, keeping the informative figure 1 and reducing figures 2*

*and 3 since there is no discussion of 1/24 and 1/48 degree resolution. The same can be said for figure 7 as 1/36 degree resolution is only briefly discussed.*

Sections 1.2 and 1.3 have been substantially edited. Figs 2, 3 and 7 and table 4 have been simplified to only include the resolutions discussed in the text. We retain 1/36 as a likely next step from the current 1/12 model (P6 L17).

*I also feel that figures 4,5 and 6 contribute little enhancement over the given text and can easily be omitted.*

Figures 4 and 5 have been replaced with the comparison between CMIP5, NNA and ORCA4 models and EN4 observations (F4, 5, 6). Old Fig 6 (new fig 7) has been retained as we feel it important include an illustration of the refinement process to help explain fig 8.

*However the treatment of other numerical schematisations is given only cursory consideration and selects, what seems to me, to be a totally arbitrary scaling factor (5) for the performance of unstructured models (in this case FVCOM is chosen) against NEMO in a limited shelf region (without any detail of these simulations), which is then applied on a global scale.*
*A more considered view of alternative numerical representations is warranted, and the authors are aware of some of these (e.g. FESOM, MPAS) and there are others (e.g. SELFE, SCHISM), and what about adaptive grid schemas?*

It is not our intention to go into the details of numerical approaches as this would require a much more comprehensive review. We now briefly describe a wider range of models (P12 L34-39) and provide some more detailed discussion on the relative efficiency of structured and unstructured mesh models (P15 L35), and also provide a more detailed consideration of what an appropriate range of scaling factors might be (P16 L5-18).

*Also, there is no discussion of the efficiency of NEMO against other comparable models which are global (e.g. MOM, HYCOM, ROMS).*

We aim at the discussion here to be model independent – the simple cost model makes no assumptions on actual model used. We now include some anecdotal estimates of present day global model turn-around time (Table 3).

*some of the processes discussed e.g. tides, sea ice, etc. will need to be included to properly represent the physical processes in these regions at this scale, so some estimates of their inclusion ought to form part of the discussion.*
We include an estimate of efficiency of a shelf sea model in Table 3. Efficiency of sea ice modelling can be a particular issues (e.g. around load balancing), but is not pertinent to the discussion here (P1 L34).

*Minor points*
Corrected

*Page 5, line 17 ": : :tides are ubiquitous in the coastal-ocean". Well, no they are not an important process to resolve everywhere, some coastal ocean regions have extremely small tides.*

Agree – we have replaced this statement with a quantification of how important tides are (by area) P4 L15

*Page 6, line 34. Please better explain the factor E*

This has been clarified P5 L23-25,

*Page 9, line 20+. Worth adding further detail regarding the sophistication of the hpg calculation and explanation of its impact on the energy cascade*
Discussion on this has been developed: P8 L37-

*Page 10, line 12. ": : : but is not generally used : : :" is it 'used' or 'required'?*
Agreed and changed P9 L34.

*Page 12, line 5. States Figure 9, should be Figure 10*
*Page 12, line 8. Need a reference for the statement "serious numerical issues"*
*Page 12, lines 13-16. Sentence badly constructed*
This has now been removed

*Page 13, line 30. What are the implications of this energy figure?*
A comment has been added P14 L29

*Page 13, line 31. The projections of compute power in Figure 9 appear to be based on extrapolations of existing architecture, how realistic is this assumption?*

It is based on an extrapolation: this is the best information we have to judge future evolution of computer resources – there are of course many unknowns. But even this uncertain information is much better than the common unquantified assertions of 'increasing computer power'.

[revised manuscript text omitted]

Font color: Auto

| Page 8: [3] Formatted | Holt, Jason T. | 06/12/2016 15:42:00 |
|---|---|---|

Font color: Auto

| Page 8: [3] Formatted | Holt, Jason T. | 06/12/2016 15:42:00 |
|---|---|---|

Font color: Auto

| Page 8: [3] Formatted | Holt, Jason T. | 06/12/2016 15:42:00 |
|---|---|---|

Font color: Auto

| Page 8: [3] Formatted | Holt, Jason T. | 06/12/2016 15:42:00 |
|---|---|---|

Font color: Auto

| Page 8: [3] Formatted | Holt, Jason T. | 06/12/2016 15:42:00 |
|---|---|---|

Font color: Auto

| Page 8: [3] Formatted | Holt, Jason T. | 06/12/2016 15:42:00 |
|---|---|---|

Font color: Auto

| Page 8: [3] Formatted | Holt, Jason T. | 06/12/2016 15:42:00 |
|---|---|---|

Font color: Auto

| Page 8: [3] Formatted | Holt, Jason T. | 06/12/2016 15:42:00 |
|---|---|---|

Font color: Auto

| Page 8: [3] Formatted | Holt, Jason T. | 06/12/2016 15:42:00 |
|---|---|---|

Font color: Auto

| Page 8: [3] Formatted | Holt, Jason T. | 06/12/2016 15:42:00 |
|---|---|---|

Font color: Auto

| Page 8: [3] Formatted | Holt, Jason T. | 06/12/2016 15:42:00 |
|---|---|---|

Font color: Auto

| Page 8: [3] Formatted | Holt, Jason T. | 06/12/2016 15:42:00 |
|---|---|---|

Font color: Auto

| Page 8: [3] Formatted | Holt, Jason T. | 06/12/2016 15:42:00 |
|---|---|---|

Font color: Auto

| Page 8: [3] Formatted | Holt, Jason T. | 06/12/2016 15:42:00 |
|---|---|---|

Font color: Auto

| Page 8: [3] Formatted | Holt, Jason T. | 06/12/2016 15:42:00 |
|---|---|---|

Font color: Auto

| Page 8: [3] Formatted | Holt, Jason T. | 06/12/2016 15:42:00 |
|---|---|---|

Font color: Auto

| Page 8: [3] Formatted | Holt, Jason T. | 06/12/2016 15:42:00 |
|---|---|---|

Font color: Auto

| Page 8: [3] Formatted | Holt, Jason T. | 06/12/2016 15:42:00 |
|---|---|---|

Font color: Auto

| Page 8: [3] Formatted | Holt, Jason T. | 06/12/2016 15:42:00 |
|---|---|---|

Font color: Auto

| Page 8: [3] Formatted | Holt, Jason T. | 06/12/2016 15:42:00 |
|---|---|---|

Font color: Auto

| Page 8: [3] Formatted | Holt, Jason T. | 06/12/2016 15:42:00 |
|---|---|---|

Font color: Auto

| Page 8: [3] Formatted | Holt, Jason T. | 06/12/2016 15:42:00 |
|---|---|---|

Font color: Auto

| Page 8: [3] Formatted | Holt, Jason T. | 06/12/2016 15:42:00 |
|---|---|---|

Font color: Auto

| Page 8: [3] Formatted | Holt, Jason T. | 06/12/2016 15:42:00 |
|---|---|---|

Font color: Auto

| Page 8: [3] Formatted | Holt, Jason T. | 06/12/2016 15:42:00 |
|---|---|---|

Font color: Auto

| Page 8: [3] Formatted | Holt, Jason T. | 06/12/2016 15:42:00 |
|---|---|---|

Font color: Auto

| Page 8: [3] Formatted | Holt, Jason T. | 06/12/2016 15:42:00 |

Font color: Auto

| Page 8: [3] Formatted | Holt, Jason T. | 06/12/2016 15:42:00 |

Font color: Auto

| Page 8: [3] Formatted | Holt, Jason T. | 06/12/2016 15:42:00 |

Font color: Auto

| Page 8: [3] Formatted | Holt, Jason T. | 06/12/2016 15:42:00 |

Font color: Auto

| Page 8: [4] Deleted | Holt, Jason T. | 23/11/2016 16:51:00 |

If it turns out that addressing these issues is problematic or computationally prohibitive

| Page 8: [4] Deleted | Holt, Jason T. | 23/11/2016 16:51:00 |

If it turns out that addressing these issues is problematic or computationally prohibitive

| Page 8: [4] Deleted | Holt, Jason T. | 23/11/2016 16:51:00 |

If it turns out that addressing these issues is problematic or computationally prohibitive

| Page 8: [4] Deleted | Holt, Jason T. | 23/11/2016 16:51:00 |

If it turns out that addressing these issues is problematic or computationally prohibitive

| Page 8: [4] Deleted | Holt, Jason T. | 23/11/2016 16:51:00 |

If it turns out that addressing these issues is problematic or computationally prohibitive

| Page 8: [5] Deleted | Holt, Jason T. | 05/12/2016 16:53:00 |

| Page 8: [5] Deleted | Holt, Jason T. | 05/12/2016 16:53:00 |

| Page 8: [5] Deleted | Holt, Jason T. | 05/12/2016 16:53:00 |

| Page 8: [5] Deleted | Holt, Jason T. | 05/12/2016 16:53:00 |

| Page 8: [6] Deleted | Holt, Jason T. | 02/12/2016 19:38:00 |

coupled ocean-atmosphere climate

| Page 8: [6] Deleted | Holt, Jason T. | 02/12/2016 19:38:00 |

coupled ocean-atmosphere climate

| Page 8: [6] Deleted | Holt, Jason T. | 02/12/2016 19:38:00 |

coupled ocean-atmosphere climate

| Page 8: [6] Deleted | Holt, Jason T. | 02/12/2016 19:38:00 |

coupled ocean-atmosphere climate

| Page 8: [6] Deleted | Holt, Jason T. | 02/12/2016 19:38:00 |

coupled ocean-atmosphere climate

| Page 8: [7] Deleted | Holt, Jason T. | 05/12/2016 20:12:00 |

of terrain following (s-)

| Page 8: [7] Deleted | Holt, Jason T. | 05/12/2016 20:12:00 |

of terrain following (s-)

| Page 8: [7] Deleted | Holt, Jason T. | 05/12/2016 20:12:00 |

of terrain following (s-)

| Page 8: [7] Deleted | Holt, Jason T. | 05/12/2016 20:12:00 |

of terrain following (s-)

| Page 12: [8] Deleted | Holt, Jason T. | 10/11/2016 16:18:00 |

Quadrilaterals in contrast have good numerical properties (particularly for wave propagation), so a conceptually attractive option would be a mixed element grid, with quadrilaterals covering the majority of the ocean and triangles used to refine the grid where needed (Figure 9). The finite volume method is readily generalised to this approach, which would have the advantage of only requiring stabilisation of numerical modes in the triangular mesh region. However, the best choice of grid arrangement (Figure 9B shows some options) is uncertain. The C-grid type arrangement is the optimal choice for wave propagation in quadrilaterals, but has serious numerical issues for the triangular mesh. Hence, any choice will be a compromise and require careful evaluation. Currently Danilov and Androsov (2015) have investigated the B-grid and Holt et al (2013) the A1 grid cases. Alternative approaches for this are global triangular or hexagonal mesh models. The former have been available for many years, but have not yet reached a level of maturity where they form the ocean component of CMIP models, whereas the latter is an emerging capability (Ringler et al., 2013).

| Page 12: [9] Deleted | Holt, Jason T. | 07/12/2016 16:28:00 |

and requires considerable investment in model configurations (a new global model must be configured and tested); so this region must be likely to endure as a focus of interest.